# WEAK-TO-STRONG GENERALIZATION WITH FAILURE TRAJECTORIES

**Ruimeng Ye[1], Zihan Wang[2], Yang Xiao[1], Zinan Ling[1], Manling Li[2], Bo Hui[1]**
[1]University of Tulsa    [2]Northwestern University
{ruy9945, bo-hui}@utulsa.edu

## ABSTRACT

Weak-to-Strong generalization (W2SG) is a new trend to elicit the full capabilities of a strong model with supervision from a weak model. While existing W2SG studies focus on simple tasks like binary classification, we extend this paradigm to complex interactive decision-making environments. Specifically, we fine-tune a strong model with trajectories of intermediate actions generated by a weak model. Motivated by the human learning process, we propose to generalize not only successful knowledge but also failure experience so that the strong model can learn from the failed trajectories accumulated by weak models. To effectively and efficiently elicit the potential of strong agents, we further construct "trajectory trees," a hierarchical representation that organizes weak model-generated action trajectories, coupled with Monte Carlo Tree Search (MCTS) to optimize the strong model. Through theoretical analysis, we provide formal guarantees for the effectiveness of our method in improving W2SG performance. Our empirical evaluations demonstrate substantial improvements in reasoning and decision-making capabilities across diverse task domains, validating the scalability and robustness of our proposed framework. Our code is available at: https://github.com/yeruimeng/TraTree.git.

## 1 INTRODUCTION

The advent of Large-scale Language Models (LLMs) has marked a significant advancement in a wide range of tasks. Currently, the alignment and supervision of these models primarily rely on human feedback and fine-tuning paradigms such as RLHF Ouyang et al. (2022); Christiano et al. (2017); Stiennon et al. (2020); Bai et al. (2022). However, as the research interest in LLMs continues growing and new capabilities of LLMs are being developed rapidly, it is believed that superintelligence (i.e., AI smarter than humans) could arrive within the next 10 years Burns et al. (2023) This raises a critical challenge, as providing reliable supervision becomes increasingly difficult when superhuman models potentially surpass human-level intelligence across numerous domains Casper et al. (2023). Thus, how to effectively supervise LLMs that may exceed human capabilities in complex tasks remains an open and pressing question.

This challenge has prompted researchers to explore alternative supervision mechanisms. A particularly promising paradigm is the Weak-to-Strong Generalization (W2SG) framework Burns et al. (2023); Ye et al. (2025), which utilizes weak models as substitutes for human supervision, eliciting strong models to learn from the weak labels generated by these less capable models. While this approach has shown remarkable performance in simple tasks such as binary classification, its application to complex scenarios such as reasoning remains largely unexplored. As the majority of existing alignment methods in LLMs leverage RLHF (reinforcement learning from human feedback) to align LLMs with human values (e.g., safety), we remark that the human values can be generalized from the weak model with the W2SG framework when reliable human supervision is unavailable. For example, if we fine-tune a weak model with SFT Ouyang et al. (2022) to follow human intention in complex decision-making, the challenge is how to generalize the intention with a weak supervisor and elicit the optimal policy in a strong model. Just like humans supervising strong models, we use weak models carrying the human's intention to align the strong models. Such a setup is called weak-to-strong learning in Burns et al. (2023). In parallel, recent research has explored performance-based

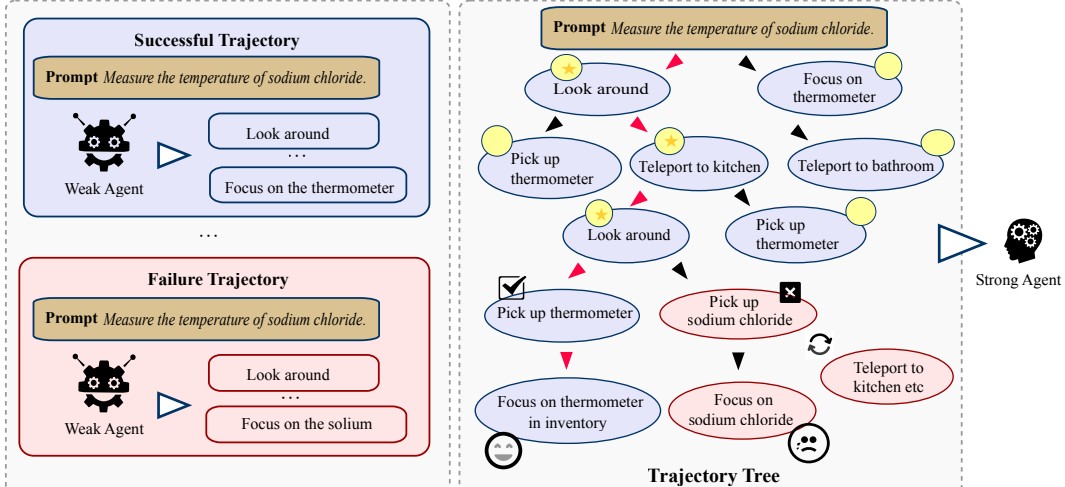

Figure 1: Illustration of the trajectory tree construction and the weak-to-strong generalization with trajectories. The left part explores different trajectories with a weak model. A trajectory tree is constructed by merging the same action path. Nodes on the tree represent different actions and performing different actions will lead to various subsequent paths. Then the trajectory tree is used to elicit the ability of the strong model.

learning approaches such as Direct Preference Optimization (DPO) Rafailov et al. (2024), which enables agents to learn from trajectory pairs during exploration. However, these approaches still face limitations due to the binary nature of pairwise preferences, which fails to capture the rich information and structural relationships among multiple reasoning paths Amini et al. (2024).

In this paper, we first extend the W2SG problem to complex decision-making tasks where the solution of an LLM agent is a trajectory of actions Li et al. (2022). Figure 1 demonstrates our proposed W2SG workflow, where we use the weak LLM agent to explore diverse trajectories in the environment, obtaining feedback, and the stronger model can learn from both positive and negative outcomes. Note that we use the weak LLM agent to explore environments to generate diverse solution trajectories, since sub-optimal solutions may limit the generalizability. The benefits are threefold: (1) The strategy addresses the limitation of entirely relying on human expert trajectories by exploring solution trajectories with a weak supervision model. It enables strong LLM agents to learn without human intervention (Section 3.1). (2) The explored trajectories can elicit the strong model to explore the larger unknown solution landscape due to the limited ability of weak models (Section 3.2); (3) The generalized knowledge can be passed with a bootstrapping framework (i.e., GPT-3→GPT-4→GPT-5). We remark that both success and failure trajectories are important for W2SG. Similar to human learning from the failure experience summarized by ancestors, the failure trajectories can elicit the strong model's ability to avoid the same failure (Section 4.2).

The cornerstone of our innovation is the trajectory trees, a hierarchical representation that fundamentally extends beyond traditional linear Chains of Thought (CoT) Wei et al. (2022). While the Tree-of-Thoughts (ToT) approach Yao et al. (2023b) also explores multiple reasoning paths, our trajectory trees explicitly organize both successful and failed trajectories from weak models, thereby capturing richer hierarchical relationships among diverse solution paths. Instead of relying on single, linear reasoning paths, we explore and collect multiple solution trajectories from weak models and construct a comprehensive tree structure. Figure 1 illustrates the construction of the trajectory tree. Different from DPO Rafailov et al. (2024) that guides learning through random contrastive preference data pairs (i.e., there is no overlap between two solution trajectories in a random preference pair), trajectory trees capture the global relationships and hierarchical structures among reasoning paths, providing more comprehensive and diverse training signals. For example, while the success trajectory (the purple path in Figure 1) and the failure trajectory (the red path) share the same prefix actions, the very first different action across two paths could be the key to the success. Compared with random pairs, such structural differences can improve the efficacy of W2SG. We traverse the trajectory tree and fine-tune the strong model with the preference of actions on the tree structure. To

further improve the performance and efficiency of W2SG, we introduce Monte Carlo Tree Search (MCTS) Grill et al. (2020); Castellini et al. (2023) as the policy optimization algorithm motivated by the success of MCTS in board games Schrittwieser et al. (2020). Optimized actions are selected by computing the cumulative reward and the node visit count. The optimization algorithms enable the strong model to efficiently generalize from policies converging to optimality with a small error probability. Specifically, the upper confidence bound applied to the trajectory trees is to deal with the exploration-exploitation dilemma. Lastly, we theoretically prove that the weak-to-strong model can surpass the strong model's performance trained on expert trajectories, even when learning from imperfect trajectories generated by the weak policy.

In the experiment, we demonstrate that weak, well-trained models can effectively produce auxiliary signals to guide the learning of strong models, establishing a more scalable pathway for improving LLM agent performance. Our primary contributions include:

- We investigate the feasibility of the weak-to-strong generalization of LLM agents in complex tasks where the solution is a trajectory of actions. Our work addresses the limitation of existing W2SG works in an analogous setup.

- We propose to construct trajectory trees to organize both success and failure trajectories explored by a weak model. Instead of relying on single reasoning paths or random contrastive pairs, the tree structure can capture the shared path between a success trajectory and a failure trajectory. The divergence between the two paths is vital for knowledge generalization.

- To the best of the author's knowledge, this is the first work that introduces MCTS in W2SG. We employ MCTS to capture hierarchical relationships between reasoning paths and provide a more detailed and complete representation compared to traditional linear CoT approaches. We also present a theoretical analysis of W2S.

- Surprisingly, we find that the W2S model can even outperform the SFT strong model in the experiment. It verifies the validity and effectiveness of our approach.

## 2 PRELIMINARY

**LLM Agent Tasks Formulation** Consider an LLM agent performing tasks formalized as a partially observable Markov decision process (POMDP) $(U, S, A, O, T, R)$. These components represent: the space of possible instructions $U$; the complete state space $S$; the set of available actions $A$; the domain of observable feedback $O$; a state transition mapping $T : S \times A \to S$; and an immediate reward function $R : S \times A \to [0, 1]$ quantifying performance for state-action pairs.

For any instruction $u \in U$, the LLM agent's policy $\pi_\theta$, parameterized by $\theta$, generates an action $a_j$ at each step $j$ sequentially according to $a_j \sim \pi_\theta(\cdot|u, a_1, o_1, \ldots, a_{j-1}, o_{j-1})$ (where $(a_1, o_1, \ldots)$ is the interaction history; for $j = 1$, history contains $u$ or an initial observation). This process yields a trajectory:

$$e = (u, a_1, o_1, \ldots, a_{n-1}, o_{n-1}, a_n, o_n), \tag{1}$$

where $n$ is the number of actions. The probability of the agent's actions in $e$ given $u$ is:

$$\pi_\theta(e|u) = \prod_{j=1}^{n} \pi_\theta(a_j|u, a_1, o_1, \ldots, a_{j-1}, o_{j-1}), \tag{2}$$

For each completed trajectory $e$, the environment assigns a final scalar score $G(e) \in [0, 1]$, which quantifies its overall quality or success rate on the task $u$. This $G(e)$ serves as the primary feedback for the trajectory. The overall performance of a policy $\pi_\theta$ is its expected score, $\mathcal{R}(\pi_\theta)$, defined as:

$$\mathcal{R}(\pi_\theta) = \mathbb{E}_{u \sim \mathcal{D}_U, e \sim \pi_\theta(\cdot|u)}[G(e)], \tag{3}$$

Here, $\mathcal{D}_U$ is a distribution over the set of initial instructions $U$. $\mathcal{R}(\pi_\theta)$ represents the policy's average task score and is our primary evaluation metric. Further details on how $G(e)$ is computed and how it relates to the immediate reward $R$ and the expected score $\mathcal{R}(\pi_\theta)$ can be found in Appendix E.

**Problem Setup** Our work investigates weak-to-strong generalization (W2SG) in the context of LLM agents performing multi-step interactive tasks. We consider two types of models: a *weak model* ($\pi_w$) and a *strong model* ($\pi_s$). These models differ in their underlying capacity (e.g., parameter count,

architecture), with $\pi_s$ assumed to have a larger capacity and potentially higher performance ceiling than $\pi_w$. We denote the base pre-trained models as $\pi_w^{\text{base}}$ and $\pi_s^{\text{base}}$. When fine-tuned on a dataset of expert-demonstrated ground truth trajectories $\mathcal{D}_{\text{expert}}$ using Supervised Fine-Tuning (SFT) Ouyang et al. (2022), these models become $\pi_w^{\text{SFT}}$ and $\pi_s^{\text{SFT}}$, respectively. The model $\pi_s^{\text{SFT}}$ serves as a crucial baseline representing a strong model trained with high-quality expert supervision.

---

**Our W2SG vs Prior Work**

This framing extends the traditional Weak-to-Strong Generalization Burns et al. (2023), which often focuses on a strong model learning from (potentially noisy) discrete labels provided by a weak supervisor. In contrast, our work applies W2SG to sequential decision-making tasks where the weak model $\pi_w^{\text{SFT}}$ generates entire trajectories of interaction. We propose that by structuring these explored experiences (via trajectory trees) and applying advanced optimization algorithms (like DPO informed by tree structure, or MCTS for path refinement), we can more effectively distill knowledge from weak model explorations to guide the strong model.

---

The open problem we address is: can supervision derived from the less capable $\pi_w^{\text{SFT}}$ (in the form of its generated trajectories) be used to elicit the full potential of $\pi_s$, potentially enabling it to achieve performance comparable to or even exceeding $\pi_s^{\text{SFT}}$? This exploration aims to determine if W2SG holds for complex interactive tasks, evaluated by the policy's expected task score $\mathcal{R}(\pi)$.

## 3 METHODS

Figure 2 depicts the workflow of our method. Our first step is to generate diverse action trajectories with a fine-tuned weak model, $\pi_w^{\text{SFT}}$. These trajectories are subsequently organized into a hierarchical trajectory tree. This tree then serves as the foundation for two proposed W2SG fine-tuning algorithms designed to enhance the strong model's ($\pi_s$) reasoning and decision-making capabilities by learning from the structured success and failure experiences of the weak model.

### 3.1 TRAJECTORY EXPLORATION

The initial step involves training the base weak LLM $\pi_w^{\text{base}}$ using SFT on an expert demonstration dataset $\mathcal{D}_{\text{expert}} = \{(X_i, Y_i)\}_{i=1}^{N_{\text{expert}}}$ to obtain $\pi_w^{\text{SFT}}$. The SFT objective is the standard negative log-likelihood:

$$\mathcal{L}_{\text{SFT}}(\theta_w) =$$
$$-\frac{1}{N_{\text{expert}}} \sum_{i=1}^{N_{\text{expert}}} \sum_{t=1}^{|Y_i|} \log \pi_w(y_t^{(i)} \mid X_i, y_{<t}^{(i)}; \theta_w), \tag{4}$$

Once $\pi_w^{\text{SFT}}$ is obtained, it is used to explore the task environments and further generalize the human feedback to the strong model. For each instruction $u$, we prompt $\pi_w^{\text{SFT}}$ multiple times, varying sampling parameters (e.g., temperature, top-p) to generate a diverse set of $M$ trajectories $\{e_1, \ldots, e_M\}$. This diversity is crucial for constructing a rich trajectory tree that captures a wide range of behaviors—including successes, failures, and suboptimal paths. The score $G(e_k)$ for each trajectory $e_k$ is provided by the environment's objective success criteria. To further enhance exploration diversity, we can optionally refine the exploration policy using a KL-divergence penalty against previously generated trajectory distributions Song et al. (2024a):

$$\mathcal{L}_{\text{explore}}(\theta_w) =$$
$$\mathcal{L}_{\text{SFT}}(\theta_w) - \lambda \cdot \text{KL}(\pi_w(\cdot|X; \theta_w) \| \pi'_{\text{explore}}(\cdot|X)), \tag{5}$$

where $\pi'_{\text{explore}}$ is a reference policy from a previous exploration phase. These collected diverse trajectories form the subsequent tree.

### 3.2 TRAJECTORY TREE CONSTRUCTION AND WEAK-TO-STRONG GENERALIZATION

The diverse trajectories collected from $\pi_w^{\text{SFT}}$ are organized into a unified trajectory tree $T = (\mathcal{V}, \mathcal{E})$, where $\mathcal{V}$ is a set of nodes and $\mathcal{E}$ represents directed edges. This tree is constructed iteratively by

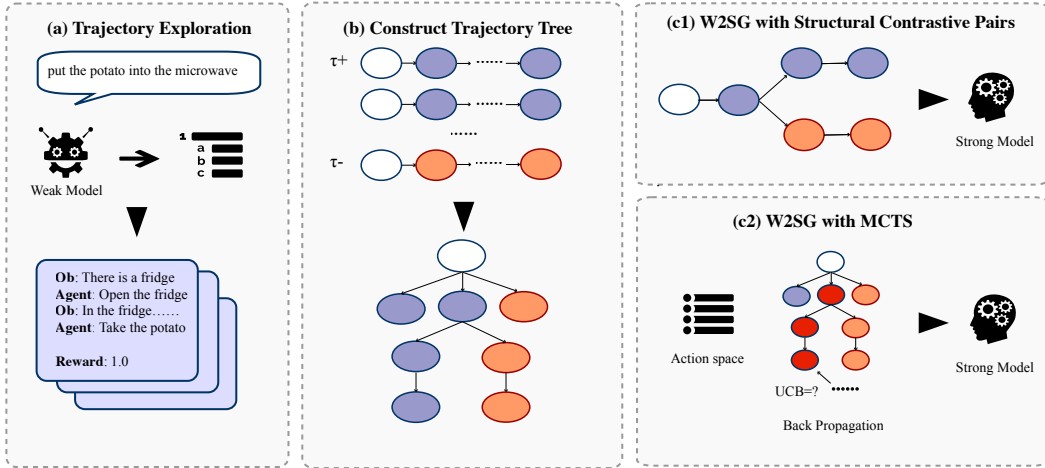

Figure 2: Illustration of the Weak-to-Strong framework. (a) Given an instruction, the weak LLM agent interacts with the environment to collect both success and failure trajectories of actions. (b) The explored trajectories are used to construct a trajectory tree by merging prefixes of actions. We propose two methods to supervise the strong model: (c1) DPO with structural contrastive failure-success pairs of trajectories instead of random pairs; (c2) Fine-tune the strong model with Monte Carlo Tree Search.

adding each explored trajectory, starting from a root node that signifies the initial instruction. Each node $v \in \mathcal{V}$ in this tree signifies an agent's "execution step" $(o_v, th_v, a_v)$, comprising the preceding environment observation $o_v$, the agent's thought $th_v$, and its subsequent action $a_v$. Edges in the tree implicitly represent the sequential progression from one such execution step to the next. To foster a compact and generalizable structure, paths are merged: when adding a new step $(o_{\text{new}}, th_{\text{new}}, a_{\text{new}})$ from a parent node, if an existing child node represents the *same action* $a_{\text{new}}$ taken from a semantically similar observation $o_v$ (where similarity between $o_v$ and $o_{\text{new}}$ is determined using sentence embeddings and a cosine similarity threshold $\xi_{\text{sim}}$), that existing child node is reused and its visit count updated. Otherwise, a new node for $(o_{\text{new}}, th_{\text{new}}, a_{\text{new}})$ is created. Our merging strategy uses exact matches for actions; consequently, even minor phrasal variations in actions lead to distinct branches. The final environment-provided score $G(e)$ for each original trajectory is associated with its terminal execution step (node) in the tree, for example, by aggregation into a `total_reward`.

A "good" trajectory tree, conducive to our W2SG methods, exhibits several key characteristics: (i) **Diversity (Breadth)**, stemming from the varied explorations of $\pi_w^{\text{SFT}}$, thereby capturing a wide range of strategies and behaviors; (ii) **Representativeness (Depth)**, with paths that are sufficiently long to represent complete or near-complete attempts at solving the given tasks; and (iii) **Informativeness**, featuring clear divergence points where different actions taken from similar (merged) states lead to demonstrably varied outcomes, as indicated by their aggregated $G(e)$ scores from the initial weak model explorations.

Based on this trajectory tree, we propose two W2SG algorithms to fine-tune the strong model: **W2SG with action preferences** Instead of directly fine-tuning with DPO Rafailov et al. (2024) based on random contrastive pairs as in common DPO practice, we propose to model the structural difference between two paths on the tree. Preference pairs are formed from divergence points within the tree, where two different continuations $\sigma^+$ and $\sigma^-$ from a shared prefix path $h$ lead to distinct aggregated $G(e)$ outcomes in the weak model's exploration. We define $\tau^+ = (h, \sigma^+)$ and $\tau^- = (h, \sigma^-)$ to emphasize these critical decision points. The strong model $\pi_s$ is then fine-tuned on the resulting dataset $\mathcal{D}_w = \{(\tau_i^+, \tau_i^-)\}_{i=1}^{N_p}$ using the following DPO loss:

$$\mathcal{L}_{\text{TreeDPO}}(\pi_s; \pi_w^{\text{SFT}}) =$$
$$- \mathbb{E}_{(\tau^+, \tau^-) \sim \mathcal{D}_w} \left[ \log \sigma \left( r_{\pi_s}(\tau^+) - r_{\pi_s}(\tau^-) \right) \right]$$
$$+ \beta \cdot \text{KL}(\pi_s \| \pi_w^{\text{SFT}}), \tag{6}$$

where $r_{\pi_s}(\tau)$ denotes the implicit DPO score of trajectory $\tau$ under the strong model $\pi_s$ (which is being optimized), and $\pi_w^{\text{SFT}}$ serves as the fixed KL reference model during this DPO training

phase. **W2SG with MCTS** While weak-to-strong generalization with the preference of actions is intuitive, the computation complexity of optimizing with all contrastive pairs on the tree is high due to the large action space on a large-scale dataset. To reduce the cost and identify informative trajectory paths within this tree structure, we use MCTS offline to search the static trajectory tree and synthesize a high-quality trajectory $e^*$ for SFT-based imitation by $\pi_s$. The MCTS process iteratively traverses the tree: child execution step nodes are selected from a parent node using UCB, which balances exploration with exploitation based on node statistics $(r_M(v), c_M(v))$. These statistics are updated via backup of terminal $G(e)$ scores from the original weak trajectories that define the traversed MCTS path. The UCB formula is:

$$\text{UCB}(v') = \frac{r_M(v')}{c_M(v')} + \gamma \sqrt{\frac{\log C_M}{c_M(v')}}, \tag{7}$$

After MCTS iterations, an optimized path $e^*$ is extracted by greedily selecting child nodes with the highest MCTS-refined average rewards $(r_M(v)/c_M(v))$ at each step. The strong model $\pi_s$ then learns from a dataset $D_{e^*}$ of these $e^*$ paths via SFT:

$$\mathcal{L}_{\text{MCTS}}(\pi_s) =$$
$$-\frac{1}{|D_{e^*}|} \sum_{e_k^* \in D_{e^*}} \sum_{t=0}^{|e_k^*|-1} \log \pi_s(a_t^{*(k)} \mid \text{context}_t^{*(k)}), \tag{8}$$

## 3.3 ANALYSIS OF W2SG

We provide a theoretical analysis for our weak-to-strong generalization (W2SG) framework, focusing on the scenario where Direct Preference Optimization (DPO) is employed with preference pairs derived from the trajectory tree. Our aim is to explain why this method, despite learning from "imperfect labels", can enable a strong model $\hat{\pi}_s^{\text{TreeDPO}}$ to surpass its SFT-trained baseline ($\pi_s^{\text{SFT}}$) and potentially recover a significant portion of the performance achieved by more heavily supervised or "ceiling" models. The policy performance metric is $\mathcal{R}(\pi)$ as defined in Section 2.

Our analysis is grounded in the Bayesian interpretation of DPO Rafailov et al. (2024); Jones et al. (1998). We consider $\pi_w^{\text{SFT}}$ as a reference policy, forming part of a prior over the strong policy $\pi_s \in \Pi_s$ (where $\Pi_s$ is the hypothesis class for strong models):

$$\log p(\pi_s) = -\beta \, \text{KL}\big(\pi_s \,\big\|\, \pi_w^{\text{SFT}}\big) + C, \tag{9}$$

The preference dataset $\mathcal{D}_w = \{(\tau_k^+, \tau_k^-)\}_{k=1}^{N_p}$ from the trajectory tree informs the likelihood, where $p(\tau_k^+ \succ \tau_k^- \mid \pi_s) = \sigma\big(r_{\pi_s}(\tau_k^+) - r_{\pi_s}(\tau_k^-)\big)$. The policy $\hat{\pi}_s^{\text{TreeDPO}}$ is then obtained by minimizing the DPO objective $\mathcal{L}_{\text{TreeDPO}}(\pi_s; \pi_w^{\text{SFT}})$ as defined in Equation equation 6 from Section 3.2. Further details on this Bayesian derivation are in Appendix F.

We introduce key assumptions for our analysis:

**Assumption 1** (Strong Model Expressivity and Superior Policy)**.** There exists a policy $\pi^* \in \Pi_s$ such that:

   1. $\mathcal{R}(\pi^*) > \mathcal{R}(\pi_s^{\text{SFT}})$, where $\mathcal{R}(\pi)$ denotes the expected score.

   2. For each $(\tau^+, \tau^-) \in \mathcal{D}_w$, we have $r_{\pi^*}(\tau^+) > r_{\pi^*}(\tau^-)$.

**Assumption 2** (Weak Model Coverage and Tree Richness)**.** The SFT-weak model $\pi_w^{\text{SFT}}$, through diverse exploration, generates trajectories yielding a variety of outcomes, with $\alpha > 0$ proportion being successful or high-quality.

**Assumption 3** (Tree-Derived Preference Informativeness)**.** Preference pairs $(\tau^+, \tau^-)$ from the trajectory tree—via shared prefixes and critical divergences identified from aggregated weak model $G(e)$ scores—are significantly more informative for DPO than unstructured, random pairs. Tree-derived pairs isolate key decision points, providing DPO with clearer, more targeted learning signals distilled from weak model experiences.

Assumption 2 concerns only coverage of the weak model's exploration and does not imply Assumption 3. Informativeness requires that divergent branches reflect outcome-relevant differences, which does not follow from coverage alone. This distinction is essential for Theorem 1. Assumption 3 formalizes the informativeness of tree-derived preferences: reducing the DPO preference loss on these

structured pairs correlates with improvements in the true return. This property is essential for triggering the loss–performance sensitivity bound in Eq. equation 13, whose derivation and justification are discussed in Appendix G.

**Theorem 1** (Performance Guarantee for W2SG via Tree-Guided DPO). $\hat{\pi}_s^{\text{TreeDPO}}$ is the policy obtained by minimizing the TreeDPO loss over the random sampling of $\mathcal{D}_w$:

$$\mathcal{R}(\hat{\pi}_s^{\text{TreeDPO}}) = \mathcal{R}(\pi_s^{\text{SFT}}) + \left(\mathcal{R}(\pi^*) - \mathcal{R}(\pi_s^{\text{SFT}})\right)$$

$$- C\sqrt{\frac{\text{KL}(\pi^*\|\pi_w^{\text{SFT}}) + \log(N_p/\delta_0)}{N_p}}, \tag{10}$$

for some constant $C > 0$. This implies that $\hat{\pi}_s^{\text{TreeDPO}}$ can outperform the SFT-strong baseline if the potential improvement margin $\mathcal{R}(\pi^*) - \mathcal{R}(\pi_s^{\text{SFT}})$ is sufficiently large relative to the third term (estimation error/complexity), which diminishes with $N_p$. Full proof details are provided in **Appendix** G. The practical interpretation of Theorem 1 is that TreeDPO is beneficial precisely when the trajectory tree generated by the weak model provides informative preference gaps along shared-prefix divergences. These structured preferences activate the loss–performance sensitivity relation, enabling the strong model to improve. When weak-model exploration collapses or the preference pairs carry no information, this relation does not activate, and TreeDPO naturally reduces to the SFT strong model without degrading its performance. This is exactly the failure-safe behavior guaranteed by the KL-regularized objective.

## 4 EXPERIMENTS

### 4.1 EXPERIMENTAL SETTINGS

**Datasets** We evaluate our approach in three environments: WebShop Yao et al. (2022), a virtual shopping platform for product search and purchase tasks; ScienceWorld Wang et al. (2022), an environment for conducting scientific experiments and analysis; and AlfWorld Shridhar et al. (2020), a household task simulation environment. WebShop and ScienceWorld provide continuous rewards between 0 and 1 based on task completion quality, while AlfWorld uses binary rewards indicating successful task completion. For each task, we use **Average Reward** and **Success Rate** in the test set as the evaluation metrics. All experiments utilize Low-Rank Adaptation (LoRA) with rank $r$ set as 64 and $\alpha$ set as 128 for efficient training. During the SFT phase, we employ the AdamW optimizer with a batch size of 32 (with gradient accumulation) and a cosine learning rate of $1e-5$. After SFT, the agent explores the training set instances to collect weak trajectories. For the ETO (Exploration Trajectory Optimization, a DPO-based baseline described later in Section 4.1) training phase, we maintain the batch size of 32 while setting the learning rate to $2e-5$. The DPO loss uses a KL penalty coefficient $\beta = 0.1$. We employ language models with different capacities to investigate weak-to-strong generalization: Llama and Qwen. By default, Llama2-7B serves as our weak model and Llama2-13B is our strong model. We also conduct experiments on the Llama3-8B, Llama2-70B and Qwen2.5. For details of the experimental setup, please refer to Appendix H.

**Baselines** To verify the effectiveness of the proposed method in W2SG, we introduce the following baselines.: 1) Standard Supervised Fine-Tuning (SFT) uses expert trajectories to conduct imitation learning. SFT is applied to both the weak and strong models on an expert demonstration dataset ($\mathcal{D}_{\text{expert}}$) to obtain the **SFT Weak Model** ($\pi_w^{\text{SFT}}$) and the **SFT Strong Model** ($\pi_s^{\text{SFT}}$). $\pi_w^{\text{SFT}}$ additionally serves as the generator of initial trajectories for our tree construction, while $\pi_s^{\text{SFT}}$ acts as a primary performance benchmark. 2) we employ **Best-of-N** ($N = 8$) **sampling**, where $\pi_s^{\text{SFT}}$ generates $N$ trajectories per task, and the one with the highest environment-provided score $G(e)$ is selected, representing a strong inference strategy. 3) **ETO (Exploration Trajectory Optimization)** Song et al. (2024b), a DPO-based method, where $\pi_s^{\text{SFT}}$ is further fine-tuned using preference pairs derived from its own environmental explorations. Moreover, to establish a high-performance mark, **a Ceiling Model** is trained by fine-tuning $\pi_s^{\text{SFT}}$ via DPO, using preference pairs that designate trajectories from $\mathcal{D}_{\text{expert}}$ as "preferred" over those generated by $\pi_s^{\text{SFT}}$'s own explorations.

### 4.2 RESULTS AND ANALYSIS

Table 1 shows the weak-to-strong generalization performance across three tasks. We can observe that the phenomenon of weak-to-strong generalization still holds for these interactive tasks. We use "W2SG" to represent the weak-to-strong performance.

| Method | WebShop | | ScienceWorld | | AlfWorld |
|---|---|---|---|---|---|
| | Avg Reward | Success Rate | Avg Reward | Success Rate | Avg Reward |
| SFT Weak Model ( Llama-2-7b+SFT) | 47.1 | 87.0 | 41.2 | 55.5 | 44.8 |
| W2SG with Tree DPO W2SG with MCTS (ours) | 53.2 **56.9** (2nd) | 97.0 **99.0** (1st) | 55.4 **58.2** (1st) | 61.1 **66.8** (1st) | 56.0 **57.5** (2nd) |
| SFT Strong Model ( Llama-2-13b+SFT) | 51.0 | 94.0 | 53.6 | 59.2 | 51.5 |
| SFT Strong Model ( Llama-2-13b+ETO) | 52.0 | 97.5 | 54.9 | 61.1 | 53.7 |
| SFT Strong Model + Best of N | 52.3 | 96.0 | 55.3 | 60.7 | 55.2 |
| Ceiling Model | **58.3** (1st) | 96.5 | **56.9** (2nd) | **63.5** (2nd) | **59.0** (1st) |

Table 1: The average reward and success rate of different approaches on three agent datasets. W2SG methods trained with weak model trajectories generalize better than strong SFT baselines across tasks, with MCTS-based W2SG achieving best performance under purely weak supervision.

| Base LLM | WebShop | SciWorld | AlfWorld |
|---|---|---|---|
| Llama3-8B+SFT | 60.8 | 59.5 | 59.7 |
| Llama3-8B+Tree DPO | 62.0 | 62.7 | 61.9 |
| Llama3-8B+MCTS | 65.3 | 67.9 | 65.7 |

Table 2: The average reward of weak-to-strong generalization to Llama3-8B.

Specifically, we can find that the fine-tuned strong model with trajectories generated by a weak model consistently outperforms the fine-tuned weak model. Compared to the SFT weak model, the W2SG with DPO improves a maximum of 4.3%. in terms of the average reward. Compared with the ceiling model based on expert trajectories, we find imperfect trajectories can effectively unlock the potential of strong models, recovering up to 39.4% of the ceiling model's performance under weak supervision. Notably, W2SG achieves these improvements without requiring additional human-annotated data, relying entirely on weak models, which is particularly important in resource-constrained scenarios.

We further verify the effectiveness of MCTS in W2SG. Surprisingly, by structuring the weak model's trajectories in a tree format and using Monte Carlo Tree Search to combine optimal nodes for generating training signals, the resulting strong model even outperforms the SFT strong model. There is an average reward improvement of 11.6% and 11.7% over the SFT strong model on WebShop and AlfWorld, respectively. Moreover, on ScienceWorld tasks, it outperforms the ceiling model trained with the ETO method. This demonstrates that the proposed method enhances weak-to-

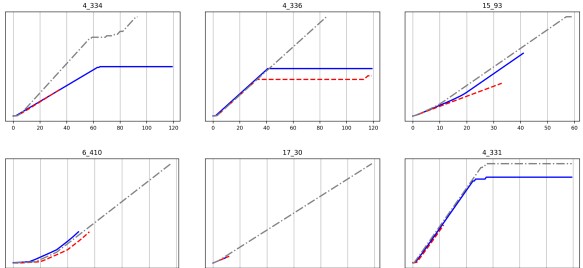

Figure 3: Three agents are compared: SFT, WTS, and MCTS. X shows the time steps while Y illustrates the scores. The task ID is shown above the plot.

strong generalization performance. We compute the p-value to judge the significance of the improvement over 5 runs with random seeds. To justify that W2SG with TreeDPO significantly outperforms the SFT strong model, we run a t-test and compute the p-value. The two-tailed P value equals 0.0003 (null hypothesis: two population means are actually equal). By conventional criteria, this difference is considered to be statistically significant. Similarly, the P value for SFT Strong and W2SG MCTS (null hypothesis: two population means are actually equal) is 0.0001. Therefore, the superiority of W2SG MCTS over the SFT strong model is also extremely statistically significant.

We also verify the phenomenon of weak-to-strong generalization with different LLM families. Table 2 shows the average reward of Llama3-8B models Supervised by Llama2-13B. The result further

verifies the feasibility of eliciting strong capabilities with weak supervision. Our work makes empirical progress on the challenge of aligning superhuman models.

**Effectiveness of Parameters.** As shown in Figure 3, we assess agent efficiency in the ScienceWorld environment by measuring how effectively an agent solves tasks with minimal action steps. Each task is broken down into fine-grained subgoals, and rewards are incrementally granted as these subgoals are achieved. Table 3 shows that when we use a higher $\beta$ value in W2SG in the ScienceWorld, the performance was inferior compared to using a lower one, which shows lower $\beta$ value helps the strong model maintain better knowledge transfer while avoiding overfitting to the reference policy.

| Method | Tree | $\beta$ | Avg Reward |
|--------|------|---------|------------|
| SFT | - | - | 53.6 |
| WTS | - | 0.1 | 54.9 |
| | - | 0.5 | 49.2 |
| MCTS | 5 | - | 57.6 |
| | 6 | - | 58.2 |
| | 7 | - | 54.9 |

Table 3: The effects of tree breadth and $\beta$.

We further explore the impact of the number of trajectories used to construct the trajectory tree in W2SG. Specifically, we adjust the decoding temperature and top-p sampling strategy of the LLM agent to collect multiple trajectories generated by the weak model. By increasing the breadth of the trajectory tree, we examined the effect of the number of collected trajectories on the results, ranging from three to ten trajectories. As shown in Figure 4, we observe that increasing the number of collected trajectories initially improves the model's performance but eventually leads to a decline. Notably, on the ScienceWorld task, the weak-to-strong model achieves performance surpassing the ceiling model when trained with six trajec-

Figure 4: The performance changes with the increase of trajectory numbers.

tories. However, this does not imply that simply adding more trajectories will consistently enhance generalization. For the AlfWorld task, when the number of trajectories used to construct the tree exceeds seven, the average reward achieved by the W2S model begins to decrease and the phenomenon is also observed in the WebShop task. While increasing the number of trajectories can improve performance, there is an optimal range beyond which additional trajectories may not yield better results and could even negatively impact W2SG.

**Performance on Qwen.** We additionally investigate the W2SG performance on the Qwen2.5 family. We used Qwen2.5-14B as the strong model and Qwen2.5-7B as the weak model. The results are shown in Table 4. TreeDPO improves the Qwen2.5-14B SFT baseline on both WebShop and AlfWorld, despite using trajectories from a significantly weaker model (Qwen2.5-7B). This trend mirrors our Llama experiments and confirms that weak-to-strong generalization is architecture-agnostic and not specific to the Llama family.

| Base LLM | Method | Webshop | Alfworld |
|----------|--------|---------|----------|
| Qwen2.5-7B | SFT | 65.5 | 58.2 |
| Qwen2.5-14B | SFT | 71.7 | 61.9 |
| Qwen2.5-14B | TreeDPO | 72.3 | 62.7 |
| Qwen2.5-14B | MCTS | 76.1 | 65.7 |

Table 4: W2SG on Qwen.

**Ablation study.** To better isolate the contribution of the tree structure, we additionally conducted an ablation on Alfworld using unstructured preference pairs on Llama3-8B, where two complete weak trajectories are randomly paired based solely on reward. Unlike tree-derived pairs, these pairs do not share a common prefix and therefore contain substantially higher noise. As shown in Table 5, the unstructured DPO variant improves slightly over the

| Method | Avg Reward |
|--------|------------|
| SFT | 59.7 |
| Unstructured DPO | 60.4 |
| TreeDPO | 61.9 |
| MCTS | 65.7 |

Table 5: Tree DPO vs. random pairs

strong SFT baseline but is noticeably weaker than TreeDPO. This confirms that shared-prefix divergences provide much clearer and more stable training signals than arbitrary preference pairs.

**Cost of trajectory exploration and MCTS Rollouts.** We remark that Trajectory tree construction is inexpensive in practice because it only processes the weak model's rollouts and does not require additional model training. For example, on Webshop, the tree construction and 100 MCTS Rollouts take 0.41 seconds and 0.23 seconds, respectively. Both components scale approximately linearly with the action horizon and the number of states in the tree, since all operations are local expansions or pointer traversals. Vocabulary size does not directly affect tree construction or search, since these stages operate purely on already-sampled trajectories rather than enumerating language actions.

**Quality of weak models.** Intuitively, the quality of the weak model will have an impact on the explored trajectories. We conduct additional experiments on Llama models and the Webshop Task with two substantially different weak models: Llama2-7B and Llama2-13B, while keeping the strong model fixed as Llama3-8B. First, as shown in Table 6, Llama2-7 B's standalone performance is far below that of Llama3-8B. Therefore, the weak model is indeed under-

| Model | Webshop | SciWorld | AlfWorld |
|-------|---------|----------|----------|
| Llama2-7B | 47.1 | 41.2 | 44.8 |
| Llama3-8B | 60.8 | 59.5 | 59.7 |

Table 6: The weakness of Llama2-7B

| Method | Llama2-7B | Llama2-13B |
|--------|-----------|------------|
| Llama3-8B+SFT | 60.8 | 60.8 |
| Llama3-8B+Tree DPO | 61.3 | 62.0 |
| Llama3-8B+MCTS | 63.7 | 65.3 |

Table 7: W2SG using Llama2-7B and Llama2-13B

performing. Second, the results in Table 7 show a clear and consistent trend: using the much weaker Llama2-7B as the weak model, the weak-to-strong update still produces a small but stable improvement over the strong SFT baseline and never leads to negative transfer. Using the stronger Llama2-13B leads to significantly larger gains. These findings confirm the monotonic relationship predicted by our theoretical analysis: weak models with low exploration capacity offer limited but non-negative signals, while stronger weak models provide more informative tree structures.

**Case Study** Figure 4 illustrates the actions of different models on the WebShop task. The strong model trained using SFT fails to complete the task correctly as it repeats incorrect actions, leading to failure and a reward score of 0. While the W2S model trained using the DPO method successfully completes the task, it makes suboptimal choices, such as selecting a product that does not fully meet the requirements. As a result, the reward score is relatively low at 0.5. However, the W2S model trained using trajectory trees not only completes the task successfully but also performs a more detailed search and makes more precise choices.

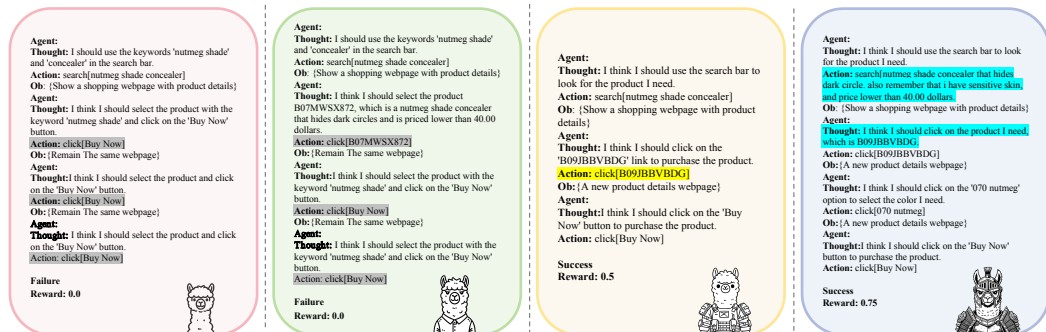

Figure 5: An example of weak-to-strong generalization in the WebShop scenario. The 4 interactions from left to right are the trajectories generated by the SFT Weak Model, the SFT Strong Model, the W2S model trained with DPO, and the W2S Model trained with MCTS.

## 5 CONCLUSION

In this work, we present a Weak-to-Strong Generalization (W2SG) framework in LLM agents. Our experiment verifies the feasibility of W2SG in reasoning and decision-making tasks. The theoretical analysis provides a robust foundation for understanding how W2SG can achieve superior performance, even when learning from imperfect trajectories. This work opens new pathways for scaling up LLM agents' training without requiring additional human supervision.

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

## A    LLM USAGE

During the preparation of this paper, we used large language models (e.g., ChatGPT) as writing assistants for language polishing and clarity improvement. The models were not involved in idea generation, experimental design, or result analysis. All scientific content and conclusions are the responsibility of the authors.

## B    REPRODUCIBILITY STATEMENT

We have made efforts to ensure the reproducibility of our results. All datasets used are publicly accessible. Anonymous source code and scripts for reproducing our experiments will be made available in the supplementary materials. These resources should allow researchers to replicate our results and extend our framework to new settings.

## C    LIMITATIONS

As discussed in Burns et al. (2023), when the weak model can contain errors that are easy to learn, the strong model could learn to imitate those errors. The limitation also exists in our setting: when a weak model is explicitly misaligned, the misalignment can also be generalized to the strong model. Due to the lack of a malicious human-labeled dataset used for RLHF, we are unable to investigate the vulnerability of the W2SG framework. The research of W2SG is still in its early stages, and there are many gaps between W2SG and other domains. Attacking through W2SG is a future work we plan to explore and strengthen our claim in various practical scenarios.

## D    RELATED WORK

**Weakly Supervised Learning.** Traditional supervised learning methodologies are undergoing significant transformation Zhou (2018); Burns et al. (2023). When it is impractical to obtain fully annotated datasets, researchers have proposed to leverage imperfect but readily available signals Lang et al. (2024); Hui et al. (2021). Weak supervision offers an alternative approach to supervised learning Sang et al. (2024); Charikar et al. (2024), aiming to keep advanced artificial intelligence models aligned with human intentions Yang et al. (2024a). A framework has recently demonstrated significant potential in enhancing Large Language Models' (LLMs) reasoning capabilities Lyu et al. (2024). Progressive frameworks enable stronger models to autonomously optimize their training data with crucial insights from weak supervisors Chen et al. (2024b;a); Kolossov et al. (2024). Recent advances have further solidified the Weak-to-Strong Generalization (W2SG) paradigm as a powerful framework for eliciting strong capabilities from imperfect supervision. Beyond empirical gains in LLM reasoning and decision-making Yang et al. (2024b); Lyu et al. (2024), recent works have explored its theoretical foundations Lang et al. (2024); Charikar et al. (2024), robustness in self-supervised settings Shin et al. (2024); Xiao et al. (2024), and methods to mitigate overfitting from noisy weak supervision in reasoning-intensive environments through staged bootstrapping Shi et al. (2025). These studies reveal that carefully designed weak signals can outperform traditional expert-labeled supervision in eliciting complex behaviors.

**LLMs Agents.** Large Language Models have evolved beyond text generation into interactive agents capable of complex reasoning and task execution Wei et al. (2022); Crispino et al. (2024); Xie et al. (2024). A fundamental breakthrough came with chain-of-thought prompting and reasoning frameworks Zhang et al. (2023); Xu et al. (2024), enabling agents to break down complex tasks into manageable steps. The field has since advanced beyond simple sequential reasoning, with researchers developing more sophisticated approaches such as Tree of Thoughts (ToT) Zhang et al. (2024b); Yao et al. (2023a) for exploring multiple reasoning paths simultaneously and Graph of Thoughts (GoT) Yao et al. (2023c); Besta et al. (2024) for handling complex problem-solving scenarios. Other works have also utilized search and value estimation for LLM reasoning. For instance, ReST-MCTS* Zhang et al. (2024a) employs a reinforced self-training approach where Monte Carlo Tree Search (MCTS) dynamically explores and expands reasoning paths to collect high-quality trajectories and infer process rewards, aiming for iterative self-improvement of a model's policy and value functions. Our weak-to-strong generalization (W2SG) framework mainly focuses on offline MCTS on a *static* trajectory tree generated by a weak model to synthesize data for fine-tuning a separate, stronger model. Further distinct are methods like Q* Wang et al. (2024) and QLASS Lin et al. (2025), which primarily focus on enhancing an LLM's performance at *inference time*. Q* learns a Q-value model as a heuristic to guide an A*-like search during the LLM's decoding process without altering the LLM's parameters. Similarly, QLASS trains a QNet to estimate step-wise Q-values from self-generated exploration trees, using this QNet to guide the agent's decision-making during inference. In contrast, our work leverages weak model explorations not for direct inference guidance, but as a source of supervision to explicitly fine-tune and transfer knowledge to a more capable strong model. However, significant challenges remain in ensuring reliability and safety, as LLM agents can exhibit inconsistent behavior. Recent work has explored multi-agent systems, where LLM agents collaborate to solve complex problems Zhang et al. (2023). The field continues to evolve rapidly, addressing challenges in agent alignment, knowledge grounding, and robustness. While Weak-to-strong generalization has been studied on reasoning ability Yang et al. (2024b), these works set up the task as a classification problem, which is different from interactive tasks in our setting.

# E  DETAILS ON TRAJECTORY SCORE AND POLICY PERFORMANCE METRIC

**Trajectory Score $G(e)$:** As defined in Section 2, $G(e)$ is the final scalar score assigned by the environment to a completed trajectory $e$ (see Equation equation 1 for trajectory definition, using your Sec 2.1 label). While the POMDP formalism includes an immediate reward function $R : S \times A \to [0, 1]$, for many complex interactive tasks such as those in WebShop or ScienceWorld, the most salient form of feedback is a terminal reward that reflects the overall success of the entire episode. This $G(e) \in [0, 1]$ directly represents this terminal reward. Formally, if one considers $R(s, a)$ to be non-zero only at the final transition leading to task completion or failure, then $G(e)$ can be seen as the sum $\sum R(s_j, a_j)$ (using notation where $s_j$ is the state after $a_j$, or $s_{j-1}$ is the state before $a_j$), which simplifies to the terminal score if intermediate rewards are zero.

**Policy Performance $\mathcal{R}(\pi_\theta)$:** The performance of a policy $\pi_\theta$, $\mathcal{R}(\pi_\theta)$, is its expected trajectory score, from Equation equation 3 (using your Sec 2.1 label): $\mathcal{R}(\pi_\theta) = \mathbb{E}_{u \sim \mathcal{D}_U, e \sim \pi_\theta(\cdot|u)}[G(e)]$. This expectation averages the trajectory scores $G(e)$ over:

1. The distribution of initial instructions or tasks $\mathcal{D}_U$.

2. The distribution of trajectories $e \sim \pi_\theta(\cdot|u)$ generated for each task $u$, accounting for any stochasticity in $\pi_\theta$ or the environment.

Thus, $\mathcal{R}(\pi_\theta)$ provides a robust measure of the policy's general effectiveness. It is the central quantity our theoretical analysis concerns and corresponds to empirical metrics like average reward over a test set.

# F  FURTHER DETAILS ON BAYESIAN INTERPRETATION OF DPO

The Direct Preference Optimization (DPO) objective, referenced in Section 3.3 as Equation equation 6 (defined in Section 3.2), is derived from finding the maximum a posteriori (MAP) estimate for the strong policy $\pi_s$ given preference data $\mathcal{D}_w = \{(\tau_k^+, \tau_k^-)\}_{k=1}^{N_p}$ and a prior $p(\pi_s)$. The posterior is $p(\pi_s \mid \mathcal{D}_w) \propto p(\mathcal{D}_w \mid \pi_s)p(\pi_s)$. The prior $p(\pi_s) \propto \exp\left(-\beta \, \mathrm{KL}\left(\pi_s \,\|\, \pi_w^{\mathrm{SFT}}\right)\right)$ (from Eq. equa-

tion 9). The likelihood $p(\mathcal{D}_w \mid \pi_s) = \prod_{k=1}^{N_p} p(\tau_k^+ \succ \tau_k^- \mid \pi_s)$, where individual preferences $p(\tau_k^+ \succ \tau_k^- \mid \pi_s) = \sigma\left(r_{\pi_s}(\tau_k^+) - r_{\pi_s}(\tau_k^-)\right)$. Minimizing the negative log-posterior directly yields the DPO loss function. This framework ensures the learned policy balances fidelity to the reference $\pi_w^{\text{SFT}}$ with exploiting preference signals from $\mathcal{D}_w$.

# G  DETAILED PROOF OUTLINE FOR THEOREM 1

This appendix provides a more detailed outline for the argument supporting Theorem 1. Let $L_{\mathcal{D}_w}(\pi_s)$ be the empirical DPO preference loss term from Equation equation 6. Let $L_{\mathcal{P}}(\pi_s)$ be its true expectation over $\mathcal{P}$, the underlying distribution of tree-derived preference pairs.

**Proof**  The DPO objective (Eq. equation 6) finds $\hat{\pi}_s^{\text{TreeDPO}}$ that minimizes empirical loss on $\mathcal{D}_w$ subject to a KL penalty. Let $L_{\mathcal{D}_w}(\pi_s)$ be the empirical preference loss term and $L_{\mathcal{P}}(\pi_s)$ be its true expectation over the distribution $\mathcal{P}$ of tree-derived preference pairs. By standard PAC-Bayesian arguments McAllester (1999), the true loss $L_{\mathcal{P}}(\hat{\pi}_s^{\text{TreeDPO}})$ can be bounded. Since $\hat{\pi}_s^{\text{TreeDPO}}$ is the minimizer of the regularized empirical loss, its regularized empirical loss is no worse than that of $\pi^*$ (from Assumption 1):

$$L_{\mathcal{D}_w}(\hat{\pi}_s^{\text{TreeDPO}}) + \beta\,\text{KL}(\hat{\pi}_s^{\text{TreeDPO}}\|\pi_w^{\text{SFT}}) \leq$$
$$L_{\mathcal{D}_w}(\pi^*) + \beta\,\text{KL}(\pi^*\|\pi_w^{\text{SFT}}), \tag{11}$$

With high probability (at least $1 - \delta_0$), concentration inequalities ensure that empirical losses are close to true expected losses for relevant policies. Thus, we can relate the true loss of $\hat{\pi}_s^{\text{TreeDPO}}$ to that of $\pi^*$:

$$L_{\mathcal{P}}(\hat{\pi}_s^{\text{TreeDPO}}) \leq L_{\mathcal{P}}(\pi^*) + \beta\,\text{KL}(\pi^*\|\pi_w^{\text{SFT}})$$
$$+ \mathcal{O}\left(\sqrt{\frac{\text{Cap}(\Pi_s, \beta\,\text{KL}) + \log(N_p/\delta_0)}{N_p}}\right), \tag{12}$$

The core step is to link this preference loss $L_{\mathcal{P}}(\pi_s)$ to the policy's actual performance $\mathcal{R}(\pi_s)$. Assumption 3 is critical here: informative, tree-derived preference pairs ensure that a lower preference loss (i.e., better alignment with true outcome distinctions) correlates strongly with higher policy performance $\mathcal{R}(\pi_s)$. We posit that for policies $\pi$ close to $\pi^*$, there's a relationship like

$$\mathcal{R}(\pi^*) - \mathcal{R}(\pi) \leq \zeta(L_{\mathcal{P}}(\pi) - L_{\mathcal{P}}(\pi^*)), \tag{13}$$

for some sensitivity $\zeta > 0$. This implies that reducing the preference error towards the optimum $L_{\mathcal{P}}(\pi^*)$ directly translates to improving $\mathcal{R}(\pi)$ towards $\mathcal{R}(\pi^*)$.

Combining the relationship equation 13 with the bound on $L_{\mathcal{P}}(\hat{\pi}_s^{\text{TreeDPO}})$ from Equation equation 12, we obtain:

$$\mathcal{R}(\hat{\pi}_s^{\text{TreeDPO}}) \geq \mathcal{R}(\pi^*) - \zeta\big(\beta\,\text{KL}(\pi^*\|\pi_w^{\text{SFT}})$$
$$+ \mathcal{O}\left(\sqrt{\frac{\text{Cap}(\Pi_s, \beta\,\text{KL}) + \log(N_p/\delta_0)}{N_p}}\right)\big), \tag{14}$$

Substituting $\mathcal{R}(\pi^*) = \mathcal{R}(\pi_s^{\text{SFT}}) + (\mathcal{R}(\pi^*) - \mathcal{R}(\pi_s^{\text{SFT}}))$ into Equation equation 14 directly leads to the form stated in Theorem 1 (Equation equation 10), where the constant $C$ in the theorem encapsulates $\zeta$ and constants from the $\mathcal{O}(\cdot)$ notation.

**1. Bounding the Empirical Loss of $\hat{\pi}_s^{\text{TreeDPO}}$**  As stated in Equation equation 11 in the main text, by definition of $\hat{\pi}_s^{\text{TreeDPO}}$ as the minimizer of the regularized empirical loss, for $\pi^*$ from Assumption 1:

$$L_{\mathcal{D}_w}(\hat{\pi}_s^{\text{TreeDPO}}) + \beta\,\text{KL}(\hat{\pi}_s^{\text{TreeDPO}}\|\pi_w^{\text{SFT}}) \leq$$
$$L_{\mathcal{D}_w}(\pi^*) + \beta\,\text{KL}(\pi^*\|\pi_w^{\text{SFT}}), \tag{15}$$

**2.  Generalization Bound (Concentration of Empirical Loss to True Loss)**  With high probability (at least $1 - \delta_0$), standard concentration inequalities (e.g., based on McAllester's PAC-Bayesian bounds McAllester (1999) or uniform convergence arguments for hypothesis class $\Pi_s$)

| Base LLM | WebShop | SciWorld | AlfWorld |
|---|---|---|---|
| Llama2-70B+SFT | 65.1 | 64.6 | 63.4 |
| Llama2-70B+Tree DPO | 65.9 | 67.4 | 65.7 |
| Llama2-70B+MCTS | 70.4 | 69.1 | 68.7 |

Table 8: The average reward of weak-to-strong generalization with Llama2-70B.

relate empirical losses to true expected losses. For any policy $\pi \in \Pi_s$: $|L_{\mathcal{P}}(\pi) - L_{\mathcal{D}_w}(\pi)| \leq$ GenError$(N_p, \delta_0, \Pi_s)$, where:

$$\text{GenError}(N_p, \delta_0, \Pi_s) =$$
$$\mathcal{O}\left(\sqrt{\frac{\text{cap}(\Pi_s, \pi_w^{\text{SFT}}) + \log(N_p/\delta_0)}{N_p}}\right), \tag{16}$$

The term $\text{cap}(\Pi_s, \pi_w^{\text{SFT}})$ represents a capacity or complexity measure of the policy class $\Pi_s$, potentially influenced by the KL regularization towards $\pi_w^{\text{SFT}}$. This term often involves quantities like VC dimension or Rademacher complexity for the functions parameterized by $\pi_s$. Applying this to $\hat{\pi}_s^{\text{TreeDPO}}$ and $\pi^*$, and combining with the result from Step 1, yields Equation equation 12 from the main text:

$$L_{\mathcal{P}}(\hat{\pi}_s^{\text{TreeDPO}}) \leq L_{\mathcal{P}}(\pi^*) + \beta \, \text{KL}(\pi^* \| \pi_w^{\text{SFT}})$$
$$+ \mathcal{O}\left(\sqrt{\frac{\text{cap}(\Pi_s, \pi_w^{\text{SFT}}) + \log(N_p/\delta_0)}{N_p}}\right), \tag{17}$$

The term $\text{KL}(\pi^* \| \pi_w^{\text{SFT}})$ here effectively captures the complexity component relevant to achieving $\pi^*$ under the prior $\pi_w^{\text{SFT}}$, as reflected in the final theorem (Eq. equation 10).

**3. Linking Preference Loss to Policy Performance $\mathcal{R}(\pi)$** This step, as described in the main text around Equation equation 13, relies on Assumption 3. The core idea is that for well-structured, informative preferences (which tree-derived pairs are assumed to be), better satisfaction of these preferences (lower $L_{\mathcal{P}}(\pi)$) implies better real-world performance $\mathcal{R}(\pi)$. The sensitivity parameter $\zeta > 0$ in $\mathcal{R}(\pi^*) - \mathcal{R}(\pi) \leq \zeta(L_{\mathcal{P}}(\pi) - L_{\mathcal{P}}(\pi^*))$ formalizes this positive correlation. The specific value of $\zeta$ depends on how directly the preferences captured by $L_{\mathcal{P}}$ (which relate to $r_{\pi_s}$ scores) translate to the overall task score $\mathcal{R}$. A strong correlation is more likely when $r_{\pi_s}$ accurately reflects true utility differences.

**4. Deriving the Final Performance Guarantee** The derivation proceeds as outlined in the main text, substituting the bound for $L_{\mathcal{P}}(\hat{\pi}_s^{\text{TreeDPO}})$ (Equation equation 12) into the loss-to-performance relationship (Equation equation 13), and then expressing $\mathcal{R}(\pi^*)$ in terms of $\mathcal{R}(\pi_s^{\text{SFT}})$ to arrive at the statement in Theorem 1 (Equation equation 10). The constant $C$ consolidates $\zeta$ and other constants arising from the $\mathcal{O}(\cdot)$ notation and the specific form of the concentration inequalities used.

# H  DATASETS

**Additional Results on Llama-2-70B**   To further validate the scalability of our method, we additionally conduct experiments with a 70B-parameter strong model. As shown in Table 8, both TreeDPO and MCTS continue to improve over the SFT baseline, with MCTS achieving the best average rewards across three tasks. This confirms that our weak-to-strong framework generalizes effectively to larger-capacity models.

**WebShop**   WebShop is a large-scale simulated e-commerce environment designed to test Large Language Model agents. It evaluates an agent's ability to understand natural language instructions, navigate web pages, and select or purchase products. The platform challenges agents with tasks such as query formulation, acting on noisy webpage text, and strategic exploration to assess their decision-making and language understanding capabilities.

**ScienceWorld**    ScienceWorld is designed to test agents' ability to perform elementary science tasks, such as conducting experiments or reasoning about scientific concepts. It evaluates an agent's scientific reasoning by requiring them to combine procedural actions with scientific knowledge to complete tasks like testing conductivity, modeling Mendelian genetics, or observing changes in states of matter.

**AlfWorld**    AlfWorld is a cross-modal framework that combines text-based and visually embodied environments to train agents in abstract reasoning and task execution. It evaluates an agent's ability to transfer abstract policies learned in textual simulations to complex embodied tasks, such as object manipulation and navigation, in visually diverse and dynamic settings.

**Experimental Settings**    Task evaluation differs across environments: WebShop and ScienceWorld implement average rewards from 0 to 1 to evaluate the model performance, whereas AlfWorld adopts a simple binary success/failure rate as metrics. We defined the task score as $100 \times$ Reward avg., which captures the average reward obtained across episodes Yao et al. (2022).

Table 9 illustrates the statistics of each dataset:

| Dataset | Train | Text-Seen | Text-Unseen |
| --- | --- | --- | --- |
| WebShop | 1,824 | 200 | - |
| ScienceWorld | 1,483 | 194 | 211 |
| AlfWorld | 3,119 | 140 | 134 |

Table 9: Dataset statistics.

Following the evaluation environment set up in Song et al. (2024b), we construct corresponding environments over three tasks to evaluate the LLM agent's performance. The original ScienceWorld and AlfWorld evaluation set comprises both seen and unseen sets. The Unseen set,s including new task instances, measures the out-of-distribution generalization of the agents. We evaluate the model performance on these unseen sets.

**Prompts**

---

**WebShop Instruction Prompts**

You are web shopping. I will give you instructions about what to do. You have to follow the instructions. Every round I will give you an observation and a list of available actions, you have to respond an action based on the state and instruction. You can use search action if search is available. You can click one of the buttons in clickables. An action should be of the following structure:
`search[keywords]`
`click[value]`
If the action is not valid, perform nothing. Keywords in search are up to you, but the value in click must be a value in the list of available actions. Remember that your keywords in search should be carefully designed.
Your response should use the following format:
Thought: I think ...
Action: click[something]

---

**ScienceWorld Instruction Prompts**

You are a helpful assistant to do some scientific experiment in an environment. In the environment, there are several rooms: kitchen, foundry, workshop, bathroom, outside, living room, bedroom, greenhouse, art studio, hallway You should explore the environment and find the items you need to complete the experiment. You can teleport to any room in one step. All containers in the environment have already been opened, you can directly get items from the containers.
The available actions are: open OBJ: open a container
close OBJ: close a container
activate OBJ: activate a device
deactivate OBJ: deactivate a device
connect OBJ to OBJ: connect electrical components
disconnect OBJ: disconnect electrical components
use OBJ [on OBJ]: use a device/item
look around: describe the current room
examine OBJ: describe an object in detail
look at OBJ: describe a container's contents
read OBJ: read a note or book
move OBJ to OBJ: move an object to a container
pick up OBJ: move an object to the inventory
pour OBJ into OBJ: pour a liquid into a container
mix OBJ: chemically mix a container
teleport to LOC: teleport to a specific room
focus on OBJ: signal intent on a task object
wait: task no action for 10 steps wait1: task no action for a step

---

**Alfworld Instruction Prompts**

Interact with a household to solve a task. Imagine you are an intelligent agent in a household environment and your target is to perform actions to complete the task goal. At the beginning of your interactions, you will be given the detailed description of the current environment and your goal to accomplish. For each of your turn, you will be given the observation of the last turn. You should first think about the current condition and plan for your future actions, and then output your action in this turn. Your output must strictly follow this format:
Thought: your thoughts.
Action: your next action. The available actions are:
1. go to recep
2. task obj from recep
3. put obj in/on recep
4. open recep
5. close recep
6. toggle obj recep
7. clean obj with recep
8. heat obj with recep

---

9. cool obj with recep where obj and recep correspond to objects and receptacles.

After your each turn, the environment will give you immediate feedback based on which you planyour next few steps. if the envrionment output "Nothing happened", that means the previous action is invalid and you should try more options.

Your response should use the following format: Thought: ¡your thoughts¿ Action: ¡your next action¿

