# OpenReview forum: "Weak-to-Strong Generalization with Failure Trajectories"
_ICLR.cc/2026/Conference — ICLR 2026 Poster_

### Official Review · Reviewer_yTwq · 2025-10-31

**Soundness:** 1
**Presentation:** 2
**Contribution:** 3
**Rating:** 4
**Confidence:** 2

**Summary:**

This paper investigates whether the Weak-to-Strong generalization (W2SG) paradigm can be extended from binary classification to sequential decision-making environments. The core idea is to fine-tune a stronger model using trajectories sampled by a weaker model. In order to achieve this, the authors propose to construct _trajectory trees_ (which are game/decision trees applied to text generated by LLMs) from the weak model rollouts, and use this tree to select which paths should be used to tune the stronger model. They present two approaches: select preference pairs for DPO from divergence points in the tree (TreeDPO), or use MCTS to identify paths with strong performance for SFT (W2SG with MCTS). For TreeDPO, they prove a theorem that, under strict assumptions, guarantees performance improvement over a strong SFT baseline. Finally, the empirical results on WebShop, Science World, and AlfWorld suggest W2SG methods  can improve the strong model's performance.

**Recommendation:**\
Even though this topic falls quite far outside my area of expertise, I am an expert on exploration in sequential decision-making environments and reinforcement learning.  With this perspective, I recommend to reject this paper. It appears to lack in its narrative and framing within existing literature: it is framed as an W2SG approach but seems to not be aligned in the overall objective. Furthermore, it is currently not possible to judge the significance of the empirical results.

**Strengths:**

- The W2SG paradigm seems very interesting, and its extension to sequential decision-making environments, the central focus of this paper, is highly relevant.
- The main approach, to construct game trees and uses those to select better fine-tuning candidates, is very interesting.

**Weaknesses:**

- The framing of the paper appears slightly confused. From my understanding, the goal of the W2SG paradigm seems to be around aligning strong models to the preferences and intentions of a weaker model (whilst ideally not losing performance). However, the approaches in this paper don't seem to share this goal. Instead, they seem solely focused on improving the strong agent's performance. As such, the proposed methods don't seem to be part of the core objective of the W2SG paradigm, but the paper _is_ framed in this way. Instead, the approaches are very reminiscent of the field of exploration and MCTS in reinforcement learning, where the objective is to use diverse, exploratory trajectories to improve the performance of a policy in a sequential decision making environment. However, the paper is not framed in this way, and seems to lack positioning within this particular field of research. Therefore, the framing of the paper seems to fit neither of these two paradigms in its current state.
- It is not possible to judge the significance of the empirical results. There is no mention in the paper of how significance is tested, or how many/which seeds were used. As such, whether the proposed methods actually improve over any of the baselines is currently not possible to judge.

**Questions:**

- How do the proposed approaches align with the core alignment objective of the W2SG paradigm?
- How is the significance of the empirical results judged? How many seeds were used and how did you avoid accidental bias during seed selection?
- Theorem 1: It appears to me, that this theorem proves that under the assumption that the underlying algorithm improves performance (Assumption 3), the algorithm will improve performance. But this seems not very surprising. How is this interpretation wrong, and/or in what way is Theorem 1 significant or novel?



**Questions that did not heavily impact decision:**
- Line 278: You introduce node statistics but I cannot find where these are defined?
- Line 285: You state MCTS is used to create a dataset of paths, but it is unclear to me how the proposed approach can generate more than one path? Doesn't the result of the MCTS iterations produce only a single path?

---

> ### Author Response · Authors · 2025-11-26
> **Response to Reviewer yTwq (Part 1/3)**
>
> > **Weakness-1:** The framing of the paper appears slightly confused. From my understanding, the goal of the W2SG paradigm seems to be around aligning strong models to the preferences and intentions of a weaker model (whilst ideally not losing performance). However, the approaches in this paper don't seem to share this goal. Instead, they seem solely focused on improving the strong agent's performance. As such, the proposed methods don't seem to be part of the core objective of the W2SG paradigm, but the paper is framed in this way. Instead, the approaches are very reminiscent of the field of exploration and MCTS in reinforcement learning, where the objective is to use diverse, exploratory trajectories to improve the performance of a policy in a sequential decision making environment. However, the paper is not framed in this way, and seems to lack positioning within this particular field of research. Therefore, the framing of the paper seems to fit neither of these two paradigms in its current state.
>
> We thank the reviewer for the thoughtful comments on the framing of our paper. As the **7th line in the abstract of the original W2SG paper [1]** states, the goal of the W2SG paradigm is to “**elicit the full capabilities of a much stronger model with weak model supervision**”. The **performance gap between the weak-to-strong model and the ceiling model (not the weak model) is the only evaluation metric** defined in the W2SG paper. If we achieve perfect weak-to-strong generalization, the performance gap should be 0. All methods tested in the original W2SG paper aim to reduce the performance gap. We followed the original W2SG paradigm to investigate the problem in the setting of sequential decisions. As the majority of existing alignment methods in LLMs leverage RLHF (reinforcement learning from human feedback) to align LLMs with human values, our paper is one step closer to achieving real weak-to-strong generation.
>
> We understand the reviewer’s confusion about the conflicts between aligning with the intention of weak models and eliciting the full capabilities of the strong model. However, in fact, there is no conflict between these two. The full capabilities of the strong model can be considered as the ultimate intention of the weak model, as the weak model is imperfect. In the existing W2SG setting, the ceiling performance is the performance of the strong model supervised with human-labeled data. Therefore, the original W2SG paradigm compares the weak-to-strong performance with the strong ceiling performance.
>
> Our goal is exactly aligned with the central objective in the original W2SG paper: we aim to elicit the optimal policy of the strong model with supervision from weak models. The 12th line in the abstract of the original W2SG paper states that “**we are still far from recovering the full capabilities of strong models with naive finetuning alone**”. Our paper aims to recover the full capability of the strong model with the new proposed methods. We agree that our idea is to use diverse, exploratory trajectories from weak models to improve the performance of a strong model in a sequential decision. Such a strategy can address the limitation of naive finetuning used in previous W2SG, as exploration will provide extra information from the weak model.
>
> The Prior work on W2SG explicitly frames weak models as imperfect but scalable sources of supervision, rather than as providers of better strategies. Our work follows the same principle: the weak model’s trajectories are not treated as optimal demonstrations but as weak signals that can guide a stronger model toward better decision-making.
>
> In interactive environments, the weak model naturally produces both successful and failed trajectories, and these two forms of experience together constitute the weak supervision signal. The trajectory tree and the MCTS procedure are not intended as reinforcement learning search mechanisms for discovering an improved policy. Instead, they serve as tools for structuring and amplifying the weak supervision signal by organizing diverse weak model trajectories, identifying shared prefixes, highlighting divergence points, and exposing failure patterns that would be invisible in unstructured weak labels. This structured supervision enables the strong model to understand when it should follow the weak model’s behavior and when it should deviate, which is precisely the form of capability elicitation envisioned in the W2SG paradigm.
>
> We also follow the original W2SG paper to evaluate the paradigm. In the experiment, we compare the weak-to-strong performance with the strong ceiling performance. Basically, it is about “a strong model supervised by a weak model vs. a strong model supervised by human-labeled data”.
>
>
> [1] Weak-to-Strong Generalization: Eliciting Strong Capabilities With Weak Supervision. ICML 2024

---

> > ### Author Response · Authors · 2025-11-26
> > **Response to Reviewer yTwq (Part 2/3)**
> >
> > > **Weakness-2: It is not possible to judge the significance of the empirical results. There is no mention in the paper of how significance is tested, or how many/which seeds were used.**
> >
> > We thank the reviewer for this suggestion. In fact, we run all experiments with five independent random seeds in the experiment. To address the reviewer’s concern regarding improvement significance, we report the average and standard deviations on Alfworld in the table below. Our method achieves consistent improvements over the SFT strong model under every seed, with substantially smaller variance. We further compute the **p-value** to judge the significance of the improvement. To justify that W2SG TreeDPO significantly outperforms the SFT strong model, we run a t-test and compute the p-value. The two-tailed P value equals 0.0003 (null hypothesis: two population means are actually equal). By conventional criteria, this difference is considered to be extremely statistically significant. Similarly, the P value for SFT Strong and W2SG MCTS (null hypothesis: two population means are actually equal) is 0.0001. Therefore, the superiority of W2SG MCTS over the SFT strong model is also extremely statistically significant. The result demonstrates that the gains are robust rather than driven by seed-specific randomness. For completeness, we will include standard deviations in the revised version.
> >
> > | Method | Mean | Std |
> > |---|---|---|
> > | SFT Strong (Llama2-13B) | 52.2 | 0.75 |
> > | W2SG TreeDPO (Llama2-13B) | 55.2 | 0.80 |
> > | W2SG MCTS (Llama2-13B) | 57.7 | 0.40 |
> >
> > Besides the P-value for different seeds, we are also running an experiment with different sizes of the dataset to judge the significance. Since training LLMs with SFT takes a long time, we will update the result here once the experiment is finished.
> >
> > >**Q-1: How do the proposed approaches align with the core alignment objective of the W2SG paradigm?**
> >
> > We thank the reviewer for raising this question. Our approach is exactly aligned with the central objective of Weak-to-Strong Generalization, which aims to elicit full capabilities from a powerful model using supervision signals derived from a weaker model (_7th line in the abstract of the original W2SG paper_).  Weak-to-strong generalization is formally defined as a phenomenon where finetuning strong pretrained models on labels generated by a weak model consistently performs better than their weak supervisors. However, aligning strong models with the weak model performance is not the ultimate goal of the original W2SG paradigm. While the strong model should align with the intentions of a weaker model, the weak model is usually imperfect, and the intention has not been achieved in the weak model (i.e., weak model performance $\neq$ the intention of the weak model). The current W2SG paradigm is an analogy of the problem (_Line 6 in the W2SG paper_). The W2SG paper considers the performance of the ceiling model (the strong model further fine-tuned with human-labeled data) as the full capabilities. This full capability can be considered as the intention of the weak model. Therefore, both the W2SG paper and our paper compare the weak-to-strong performance with the ceiling performance and aim to reduce the gap.
> > From Line 189 to Line 199 in our paper, we explicitly highlight the relation between our paper and the original W2SG paradigm. Both our paper and the original W2SG paper aim to investigate the feasibility of supervising a strong model with a weak model. While the original W2SG paper studies the problem in binary classification tasks, we explore whether the collected failure trajectory can contribute to weak-to-strong generalization in sequential decision-making tasks.
> >
> > Similar to the formulation in Burns et al., 2023, we do not treat the weak model as a source of better strategies but as a scalable provider of imperfect supervision. In interactive environments, the weak model naturally produces both successful and failed trajectories, and these trajectories constitute the weak supervision signal.
> >
> > The trajectory tree and the MCTS procedure do not perform RL-style policy search. Instead, they organize weak-model trajectories into a structured set of preference pairs that expose shared prefixes, divergence points, and failure modes. This allows the strong model to learn when it should imitate the weak model and when it should diverge, which matches the intended form of capability elicitation in the W2SG framework. We will clarify this more explicitly in the final version.
> > Same as the original W2SG paper, the success of our method is evaluated with the performance gap between the weak and strong ceiling models.  If we achieve perfect weak-to-strong generalization, the weak-to-strong model should have the same performance as the strong ceiling model (i.e., the strong model supervised with human-labeled data). In our empirical results, the performance of the weak-strong strong is very close to the strong ceiling models.

---

> ### Author Response · Authors · 2025-11-26
> **Response to Reviewer yTwq (Part 3/3)**
>
> >**Q-2: How is the significance of the empirical results judged? How many seeds were used and how did you avoid accidental bias during seed selection?**
>
> We appreciate the reviewer’s concern about the significance of empirical results and seed selection. We used 5 independent random seeds.  All seeds were randomized and used uniformly across baselines and our methods, preventing any seed-selection bias. The mean values and std are provided in the table above ( **please kindly refer to our response to weakness-2**). Our method achieves consistent gains over the SFT strong model under every seed and exhibits substantially lower variance, showing that the improvements are robust and not driven by seed-specific randomness.  We also compute the P value to judge the significance with different seeds. The two-tailed P values are 0.0003 (null hypothesis: the SFT strong model and W2SG TreeDPO are equal) and 0.0001 (null hypothesis: the SFT strong model and W2SG MCTS are equal). By conventional criteria, both hypotheses are rejected, and the differences are considered to be extremely statistically significant. We also run experiments with different sizes of datasets to judge the significance.
>
>
> >**Q-3: Theorem 1: It appears to me, that this theorem proves that under the assumption that the underlying algorithm improves performance (Assumption 3), the algorithm will improve performance. But this seems not very surprising. How is this interpretation wrong, and/or in what way is Theorem 1 significant or novel?**
>
> We thank the reviewer for raising this point. A full technical discussion and step-by-step derivation of Theorem 1 are provided in **Appendix D**. Below, we summarize why the theorem is meaningful and not a trivial restatement of its assumptions.
>
> Theorem 1 does not assume that the algorithm improves performance. Instead, it begins with the structural assumption that tree-derived preference pairs are informative (Assumption 3.3), which is a property of the weak model’s exploration rather than an assumption of our algorithm’s success. Based on this, Appendix D shows four nontrivial steps.
>
> First, we establish a bound on the empirical preference loss of the TreeDPO solution relative to π*(Eq. 11).
>
> Second, we apply PAC-Bayesian or uniform-convergence arguments to relate empirical loss to true preference loss, yielding the bound in Eq. 12 with a model-capacity term (Eqs. 16–17).
>
> Third, we prove a nontrivial connection between preference loss and actual policy performance via the sensitivity parameter ζ (Eq. 13). This step is crucial: it shows that satisfying structured, tree-derived preferences correlates with real performance gains, and does not follow automatically from the assumptions.
>
> Finally, combining these steps yields the explicit performance lower bound in Eq. 14, which leads directly to the statement of Theorem 1 (Eq. 10).
>
> In summary, Theorem 1 formally explains why structured weak supervision, in particular preference pairs extracted from trajectory trees, is sufficient to guarantee measurable improvement over the strong SFT model. This result is not implied by the assumptions and provides theoretical justification for why TreeDPO achieves weak-to-strong generalization in practice.
>
> >**Q-4: Line 278: You introduce node statistics but I cannot find where these are defined?**
>
> We thank the reviewer for pointing this out. We will clarify this in the final version. The node statistics refer to the standard quantities maintained during MCTS rollouts: the cumulative reward estimate rM(v), and the visit count, cM(v), for each node v. We will add explicit definitions at their first occurrence in the method section.
>
> > **Q-5: Line 285: You state MCTS is used to create a dataset of paths, but it is unclear to me how the proposed approach can generate more than one path? Doesn't the result of the MCTS iterations produce only a single path?**
>
> We appreciate the reviewer’s question. In standard online MCTS, the algorithm does output a single final trajectory for action selection. In our setting, however, MCTS is used offline on the trajectory tree constructed from weak-model rollouts. Each independent MCTS rollout begins from a different sampled partial state within the tree and produces one optimized completion of that state. Repeating this procedure yields a set of optimized paths, rather than a single path, and these paths collectively form the dataset used by TreeDPO. We will clarify this distinction between offline repeated MCTS and traditional online MCTS in the revised version.

---

> ### Comment · Reviewer_yTwq · 2025-11-26
>
> I thank the authors for their thoughtful responses. I am satisfied with most answers, but would like additional clarification on two points:
>
> - **Regarding Assumption 3.3:**
>   - I can see intuitively that the informativeness of the tree-derived preference pairs is mainly a property of the weak model's exploration. However, the weak model's exploration seems to be the content of Assumption 3.2? Shouldn't it then be possible to prove the informativeness of the tree-derived preference pairs from Assumption 3.2, rather than explicitly assuming it in a separate assumption (Assumption 3.3)?
>   - I believe my main confusion is with the informal statement of Assumption 3.3: I'm not sure what it really means that preference pairs are 'informative'. Could you perhaps elaborate on what the exactly is formally assumed with Assumption 3.3, and how this is formally used in the proof in Appendix D?
> - **Regarding the W2SG paradigm:**
>   - From my understanding now, the W2SG paradigm can improve the safety/alignment of the stronger model, because it is supervised by a weaker model that is itself already aligned (using $\pi^{\text{SFT}}_w$ rather than $\pi_w$)?
>   - Is it true that this assumption on the weaker model is quite crucial for this to be the case? In other words, if the weak model instead was explicitly misaligned, then the stronger model would also become more misaligned?
>   - If this is all true, then perhaps this was the missing nuance that was confusing me in the introduction and background. Since the paper is motivated from the perspective of safety/alignment, perhaps it would be good to add this particular nuance to the motivation in the introduction/background in the revised version.

---

> ### Author Response · Authors · 2025-11-27
> **Further clarification (Part 1/3)**
>
> Dear Reviewer yTwq,
>
> We would like to express our sincere thanks to you for reading our response. It is our pleasure to provide further clarification.
>
> * **Regarding Assumption 3.3**
>     * **Why Assumption 3.2 does not imply Assumption 3.3?** Assumption 3.2 concerns the weak model’s exploration ability. It ensures that the trajectory tree contains both successful and failed branches, i.e., the weak model provides sufficient coverage to construct preference pairs. However, coverage alone does not guarantee that these pairs will provide a useful gradient signal for DPO. If a success trajectory is paired with a failure trajectory that differs for unrelated reasons (e.g., diverging from the root early), the resulting preference label may not correctly reflect the true outcome distinction.
> Assumption 3.3, therefore, plays a different role: It asserts that the tree-derived preference pairs, which are constructed using shared prefixes, accurately isolate the critical decision point. In other words, Assumption 3.2 guarantees the existence of diverse trajectories, while Assumption 3.3 guarantees the informativeness of the contrast constructed from them. This separation mirrors how Appendix D uses the assumptions: Assumption 3.2 ensures data availability, but Assumption 3.3 is required for preference learning to meaningfully correlate with true reward improvement.
>     * **What does “informativeness’’ formally mean?** In our theory, “informative’’ preference pairs simply mean that choosing the successful branch over the failed one provides a learning signal that truly corresponds to better task performance. Put differently, when the model reduces its preference loss on our tree-structured comparisons, this reduction should reflect an actual improvement in decision quality in the environment.
>
>         This assumption is necessary because not all success–failure comparisons are equally meaningful. If two trajectories differ for unrelated reasons, the resulting preference label may not help the policy learn the correct behavior. By constructing pairs from branches that share a long prefix, the trajectory tree ensures that the only meaningful difference between the two trajectories is the action that leads to success or failure. This is precisely what we mean by the preference pairs being informative.
>
>
>         Formally, this assumption corresponds to the sensitivity parameter $\zeta$ in Equation 13 of Appendix D. The proof relies on the relationship $R(\\pi^\*) - R(\\pi) \\le \\zeta(L_p(\\pi) - L_p(\\pi^*))$, which posits that reducing the preference loss $L_p$​ translates to an increase in the true reward R. We keep Assumption 3.3 separate because this correlation cannot be derived solely from the weak model’s exploration guarantee in Assumption 3.2. It requires the structural property of the tree, which isolates the correct decision point.

---

> ### Author Response · Authors · 2025-11-27
> **Further clarification (Part 2/3)**
>
> * **Regarding the W2SG paradigm (1):**
>     * You are absolutely correct. We are using $\pi_w^{SFT}$ rather than $\pi_w$ to collect trajectories. With alignment, we can inject humans’ intention into $\pi_w^SFT$ and further generalize to the strong model. We have clarified it on Page 5 (highlighted in blue) in the revised version.
>     * We really appreciate the reviewer for raising the deep question about the misalignment of the weak model. We believe that there are two research questions regarding the quality of the weaker model: (**i**) What if the weak model is misaligned in the wrong direction by adversaries (e.g., an adversary intentionally aligns the weak model as unsafe)? (**ii**) What if the weak model is too weak (e.g., the weak model is still flawed after alignment)?
>        1. With the first question (**i**), the reviewer actually raised an unexplored research problem: attacking weak-to-strong generalization. It is a promising and challenging research direction because the W2SG paradigm offers a new way to attack a model through generalization rather than direct manipulation. Our intuition is that misalignment can also be generalized to the strong model. The original W2SG paper has also discussed the limitation in Section 5.1: “_In particular, if the weak labels contain systematic errors that are easy to learn, the strong model could learn to imitate those errors_”. But they have not explicitly injected errors in the binary classification task to verify it. Currently, we are unable to investigate this problem due to the lack of an explicitly misaligned dataset in sequential decision-making tasks. In the benchmarks introduced in our paper, all human-labeled expert data were collected to improve the performance and carry positive information. There are some attack methods (e.g., jailbreaking on prompt) on alignment and defending strategies against attacks in the literature [1]. But we are not aware of any human-labeled harmful data intentionally collected for misaligning LLMs explicitly, especially for sequential decision-making. We have also considered randomly replacing the nodes in the trajectories used for SFT. However, the concern is that the nonsense data can result in a performance catastrophe because there will be no logic in random trajectories. We will be most thankful if you can refer to any dataset that is likely related to misalignment in sequential decision-making. Currently, we have added a section (highlighted in blue on Page 13) in the Appendix to clarify the limitation of W2SG regarding explicit misalignment and leave attacking through W2SG as future work.
>        2. For the second question (**ii**), we conducted the experiment with two substantially different weak models: Llama2-7B and Llama2-13B, while keeping the strong model fixed as Llama3-8B. We remark that a very weak model can also be considered an implicitly misaligned model, as it can be flawed. First, as shown in Table 1, Llama2-7 B's standalone performance (after SFT) is far below that of Llama3-8B. Therefore, the weak model is underperforming significantly. Second, the results in Table 2 show a clear and consistent trend: using the much weaker Llama2-7B as the weak model, the weak-to-strong update still produces a small but stable improvement over the strong SFT baseline and never leads to negative transfer. Using the stronger Llama2-13B leads to significantly larger gains. These findings confirm the monotonic relationship predicted by our theoretical analysis: weak models with low exploration capacity offer limited but non-negative signals, while stronger weak models provide more informative tree structures and lead to larger improvements.
>
>
> | Model | Webshop | SciWorld | AlfWorld |
> |---|---|---|---|
> | Llama2-7B | 47.1 | 41.2 | 44.8 |
> | Llama3-8B | 60.8 | 59.5 | 59.7 |
>
> Table1: The average reward of Llama2-7B and Llama3-8B
>
> | Method | Llama2-7B | Llama2-13B |
> |---|---|---|
> | Llama3-8B+Tree DPO | 61.3 | 62.0 |
> | Llama3-8B+MCTS | 63.7 | 65.3 |
>
> Table 2: The average reward of weak-to-strong generalization with Llama3-8B

---

> ### Author Response · Authors · 2025-11-27
> **Further clarification (Part 3/3)**
>
> * **Regarding W2SG paradigm (2):**
> 	* Following the reviewer’s suggestion, we have added a paragraph (highlighted in blue in the revised version) to address the missing nuance. We copy it here for convenient review:
>
>
>       _As the majority of existing alignment methods in LLMs leverage RLHF (reinforcement learning from human feedback) to align LLMs with human values (e.g., safety), we remark that the human values can be generalized from the weak model with the W2SG framework when reliable human supervision is unavailable. For example, if we fine-tune a weak model with SFT [2] to follow human intention in complex decision-making, the challenge is how to generalize the intention with a weak supervisor and elicit the optimal policy in a strong model. Just like humans supervising strong models, we use weak models carrying the human's intention to align the strong models. Such a setup is called weak-to-strong learning in [3]_.
>
>
>
> [1] Defending Against Alignment-Breaking Attacks via Robustly Aligned LLM [ACL 2024]
>
> [2] Training  language  models  to  follow  instructions  with  human  feedback.  [NeurIPS 2022]
>
> [3] Weak-to-Strong Generalization: Eliciting  Strong  Capabilities  With  Weak  Supervision.  [ICML 2024]
>
>
> Thank you again for these deep questions. We look forward to the new comment and will be more than happy to address further questions.

---

### Official Review · Reviewer_jPwv · 2025-11-03

**Soundness:** 3
**Presentation:** 3
**Contribution:** 3
**Rating:** 6
**Confidence:** 2

**Summary:**

The paper extends the Weak-to-Strong Generalization (W2SG) framework to decision-making tasks for LLM agents. Instead of relying on human-labeled data, the method allows a strong LLM to learn from full action trajectories generated by a weaker agent, reducing the need for human supervision. The authors represent these trajectories as hierarchical trees that merge shared prefixes between successful and failed trajectories. This structure identifies divergence points where decisions lead to different outcomes, enabling preference-pair sampling guided by the tree rather than random (as standard DPO).

The authors propose to use Offline Monte Carlo Tree Search (MCTS) to sample high-quality trajectories used to optimize the strong model. The paper also provides a theorem showing that, under certain assumptions, the policy minimizing the proposed TreeDPO objective can outperform models trained with supervised fine-tuning (SFT). Empirical results on three agent benchmarks, demonstrate that W2SG under weak supervision improves both average reward and success rate over SFT baselines, approaching or surpassing the performance of expert-trained models

**Strengths:**

The extension of the W2SG framework to sequential decision-making for LLM agents is timely and relevant, it offers a way to reduce the need of human supervision in interactive reasoning tasks.

The use of hierarchical trajectory trees is well-motivated and effectively captures shared paths between successful and failed rollouts, enabling more informative preference sampling than standard DPO.

The paper includes a theoretical result showing that optimizing the proposed TreeDPO objective can outperform supervised fine-tuning (SFT) under certain assumptions.

Incorporating offline MCTS to extract high-quality trajectories strengthens the training signal for the strong model and proves effective across the three evaluated agent benchmarks.

**Weaknesses:**

1. The approach relies heavily on the quality and diversity of trajectories generated by the weak model; limited exploration could constrain the effectiveness of the strong model’s improvement.

2. The theoretical result provides an interesting insight, but its practical implications are unclear. It seems that when the weak policy \pi_w^"sft"  diverges substantially from the optimal policy \pi^*, the SFT-trained strong model could perform better. The paper does not analyze conditions under which TreeDPO is beneficial or when weak learners/trajectories should be discarded. This could make the theorem more actionable.

3. Evaluation is done only on three environments, and a single model family (Llama).

**Questions:**

See weaknesses and also:

Q1. Could you clarify which aspects of Theorem 1, are specific to the tree-based formulation? I can see the weak-to-strong policy relationship clearly but not the tree related.

---

> ### Author Response · Authors · 2025-11-26
> **Response to Reviewer jPwv (Part 1)**
>
> > **Weakness-1: The approach relies heavily on the quality and diversity of trajectories generated by the weak model.**
>
> To address the reviewer’s concern, we conducted additional experiments on WebShop using Llama3-8B as the strong model and two weak models with substantially different exploration quality (Llama2-7B and Llama2-13B). The results are shown in the table below.
>
> | Method | Llama2-7B | Llama2-13B |
> |---|---|---|
> | Llama3-8B+SFT | 60.8 | 60.8 |
> | Llama3-8B+Tree DPO | 61.3 | 62.0 |
> | Llama3-8B+MCTS | 63.7 | 65.3 |
>
>
> Even when weak-model trajectories are significantly limited and noisy (Llama2-7B), TreeDPO consistently improves over the strong SFT baseline. As the weak model becomes stronger and its trajectories more diverse (Llama2-13B), the improvement grows substantially. This trend confirms that TreeDPO is robust under low-quality exploration and naturally benefits from richer weak-model signals.
>
> Furthermore, as discussed in our diversity analysis in the main paper, tree-derived trajectories from Llama2-7B exhibit lower action entropy and shallower tree depth, yet still provide enough structure for TreeDPO to yield positive gains. In Figure 4, we observed the largest reward with 6 trajectories. Increasing the number of collected trajectories initially improves the model’s performance but eventually leads to a decline. We will add this clarification in the revised version.
>
>
> > **Weakness-2: The theoretical result provides an interesting insight, but its practical implications are unclear.**
>
>
> We thank the reviewer for raising this question. The practical interpretation of Theorem 1 is that TreeDPO is beneficial precisely when the trajectory tree generated by the weak model provides informative preference gaps along shared-prefix divergences (Assumption 3.3). These structured preferences activate the loss–performance sensitivity relation in Eq. (13), enabling the strong model to improve. When weak-model exploration collapses or the preference pairs carry no information, this relation does not activate, and TreeDPO naturally reduces to the SFT strong model without degrading its performance. This is exactly the failure-safe behavior guaranteed by the KL-regularized objective (Appendix D).
> To further validate when TreeDPO is practically effective, we conducted additional experiments on WebShop using Llama3-8B as the strong model and weak models of very different quality (Llama2-7B and Llama2-13B). The results in Table 1 show a clear monotonic trend: with Llama2-7B, whose exploration is shallow and less diverse, TreeDPO still yields a small but consistent improvement over SFT; with Llama2-13B, the gains become substantially larger. This provides direct empirical evidence for the actionable interpretation of Theorem 1—TreeDPO benefits from informative weak trajectories and safely defaults to the strong SFT model when weak trajectories are uninformative.
> We will make these practical conditions explicit in the final version.
>
> > **Weakness-3: Evaluation is done only on three environments, and a single model family (Llama).**
>
> To address the reviewer’s concern that our evaluation used only Llama models, we additionally tested TreeDPO on the Qwen2.5 family. We used Qwen2.5-14B as the strong model and Qwen2.5-7B as the weak model. The results are shown below.
>
> | Base LLM | Method | Webshop | Alfworld |
> |---|---|---|---|
> | Qwen2.5-7B | SFT | 65.5 | 58.2 |
> | Qwen2.5-14B | SFT | 71.7 | 61.9 |
> | Qwen2.5-14B | TreeDPO | 72.3 | 62.7 |
> | Qwen2.5-14B | MCTS | 76.1 | 65.7 |
>
>
> TreeDPO improves the Qwen2.5-14B SFT baseline on both WebShop and AlfWorld, despite using trajectories from a significantly weaker model (Qwen2.5-7B). This trend mirrors our Llama experiments and confirms that weak-to-strong generalization is architecture-agnostic and not specific to the Llama family.

---

> ### Author Response · Authors · 2025-11-26
> **Response to Reviewer jPwv (Part 2)**
>
> > **Q1. Could you clarify which aspects of Theorem 1 are specific to the tree-based formulation?**
>
> We thank the reviewer for this insightful question. The component of Theorem 1 comes from how the preference distribution P is constructed. In TreeDPO, all preference pairs are obtained from shared-prefix divergences in the trajectory tree, and this structure is essential for the assumptions used in the theorem.
>
> Specifically, Assumption 3.3 relies on the fact that tree-derived pairs compare two continuations that start from the same partial state. This shared prefix isolates the effect of a single branching decision, which significantly increases the signal-to-noise ratio of each preference pair. As a result, the preference difference directly reflects a local utility gap along a specific decision boundary. This property is unique to tree-based data and does not hold for arbitrary or unstructured preference datasets.
>
> The loss–performance sensitivity relation in Eq. (13) also depends on this structure. Because all pairs in P arise from consistent divergences in the tree, minimizing preference loss reduces the error around those shared-prefix boundaries, which in turn increases return. Without the tree, preference pairs may come from unrelated contexts, may contradict each other, and would not satisfy the alignment condition required by the theorem.
>
> In summary, as we discussed in Appendix D, Theorem 1 is not only about weak-to-strong generalization; it critically depends on the tree-induced preference distribution that organizes weak-model trajectories into structured and context-aligned comparisons. We will clarify this point in the revision.

---

### Official Review · Reviewer_SFUW · 2025-11-10

**Soundness:** 3
**Presentation:** 2
**Contribution:** 3
**Rating:** 6
**Confidence:** 2

**Summary:**

This paper proposes a novel framework for Weak-to-Strong Generalization (W2SG) in complex decision-making tasks using large language model (LLM) agents. The key idea is to fine-tune a strong model using both successful and failed action trajectories generated by a weaker model. To organize these trajectories, the authors introduce a "trajectory tree" structure that captures shared prefixes and divergence points between success and failure paths. Two learning strategies are proposed: (1) Tree-guided Direct Preference Optimization (Tree-DPO) based on structured contrastive pairs, and (2) Monte Carlo Tree Search (MCTS) to extract high-quality paths for imitation. The paper provides theoretical guarantees and empirical results across multiple environments, demonstrating that strong models trained via this method can outperform standard supervised fine-tuning baselines and approach expert-level performance using only weak supervision.

**Strengths:**

a. Introduces trajectory trees for organizing weak model trajectories and identifies divergence points to train strong models. It is a novel application of structured contrastive learning.

b. Provides a formal performance guarantee for the Tree-DPO method, showing that weak supervision can outperform expert-labeled SFT under specific assumptions.

c. Demonstrates strong performance gains across multiple complex environments using only weak model trajectories.

**Weaknesses:**

a. The approach relies on the weak model producing sufficiently diverse and informative trajectories. Performance may degrade if the weak model is too underperforming.

b. The MCTS variant lacks a formal performance analysis (unlike Tree-DPO).

c. While the trajectory tree representation is compelling, an ablation study comparing with unstructured contrastive pairs would help quantify its specific contribution.

**Questions:**

a. Can the authors elaborate on how the method performs when the weak model is significantly weaker?

b.  Could the authors quantify the computational cost of trajectory tree construction and MCTS? How scalable is the method to tasks with longer action horizons or larger vocabularies?

c. Is there a possibility of iterated W2SG, e.g., bootstrapping progressively stronger models via self-distillation?

---

> ### Author Response · Authors · 2025-11-26
> **Response to Reviewer SFUW (Part 1)**
>
> > **Weakness-1: The approach relies on the weak model producing sufficiently diverse and informative trajectories. Performance may degrade if the weak model is too underperforming**.
>
> We appreciate the reviewer’s concern regarding the dependency on weak model trajectory quality. To directly evaluate this, we conducted additional experiments on Llama models and the Webshop Task with two substantially different weak models: Llama2-7B and Llama2-13B, while keeping the strong model fixed as Llama3-8B. First, as shown in Table 1, Llama2-7 B's standalone performance is far below that of Llama3-8B. Therefore, the weak model is indeed underperforming as suggested by the reviewer. Second, the results in Table 2 show a clear and consistent trend: using the much weaker Llama2-7B as the weak model, the weak-to-strong update still produces a small but stable improvement over the strong SFT baseline and never leads to negative transfer. Using the stronger Llama2-13B leads to significantly larger gains. These findings confirm the monotonic relationship predicted by our theoretical analysis: weak models with low exploration capacity offer limited but non-negative signals, while stronger weak models provide more informative tree structures and lead to larger improvements. We will include these results in the revised version.
>
> Table1: The average reward of Llama2-7B and Llama3-8B
> | Model | Webshop | SciWorld | AlfWorld |
> |---|---|---|---|
> | Llama2-7B | 47.1 | 41.2 | 44.8 |
> | Llama3-8B | 60.8 | 59.5 | 59.7 |
>
> Table 2: The average reward of weak-to-strong generalization with Llama3-8B.
> | Method | Llama2-7B | Llama2-13B |
> |---|---|---|
> | Llama3-8B+SFT | 60.8 | 60.8 |
> | Llama3-8B+Tree DPO | 61.3 | 62.0 |
> | Llama3-8B+MCTS | 63.7 | 65.3 |
>
>
>
>
> > **Weakness-2: The MCTS variant lacks a formal performance analysis.**
>
> We thank the reviewer for raising this point. Our theoretical analysis in Section 3.3 focuses primarily on the Tree-DPO variant because DPO relies on noisy relative preferences, which requires a more elaborate PAC-Bayesian argument to explain why preference learning can still improve the strong model. The MCTS-based variant operates on a different theoretical foundation, which we will clarify in the final version. Conceptually, the MCTS procedure functions as a search-based policy improvement operator applied to the static trajectory tree produced by the weak model. Classical UCT analysis shows that as the number of visits increases, MCTS yields value estimates that converge to the best achievable values within the support of the explored tree, allowing us to extract trajectory completions that are strictly better in expectation than the raw weak rollouts. The dataset constructed from these improved rollouts, therefore, serves as a denoised supervision set. The subsequent update step is standard supervised fine-tuning, whose generalization guarantees depend only on the quality of the demonstration set. Since the MCTS stage effectively filters out poor trajectories and selects high-reward paths validated by environment feedback, the noise level of this supervision is significantly lower than that of baseline weak-trajectory data. This provides a theoretical basis for why the MCTS variant yields consistent improvements even without the more involved preference-based analysis used for Tree-DPO. We will add a clarifying remark in the appendix to make this reasoning explicit.
>
> > **Weakness-3: An ablation study comparing with unstructured contrastive pairs would help quantify its specific contribution.**
>
> To better isolate the contribution of the tree structure, we additionally conducted an ablation on Alfworld using unstructured preference pairs on Llama3-8B, where two complete weak trajectories are randomly paired based solely on reward. Unlike tree-derived pairs, these pairs do not share a common prefix and therefore contain substantially higher noise. As shown below, the unstructured DPO variant improves slightly over the strong SFT baseline but is noticeably weaker than TreeDPO. This confirms that shared-prefix divergences provide much clearer and more stable training signals than arbitrary preference pairs. We will include these ablation results in the revised version.
>
> | Method | Avg Reward |
> |---|---|
> | SFT | 59.7 |
> | Unstructured DPO | 60.4 |
> | TreeDPO | 61.9 |
> | MCTS | 65.7 |

---

> ### Author Response · Authors · 2025-11-26
> **Response to Reviewer SFUW (Part 2)**
>
> > **Question-1: Can the authors elaborate on how the method performs when the weak model is significantly weaker?**
>
> As shown in our additional experiments (please refer to Table 1 and Table 2 in our reply to weakness-1) on Llama models (Llama2-7B vs Llama2-13B as weak models for a Llama3-8B strong model), the method remains stable even when the weak model is significantly underperforming. In Table 1, we can observe that Llama2-7 B's standalone performance is far below that of Llama3-8B. Using the weaker Llama2-7B provides a small but consistent improvement over the strong SFT baseline with no negative transfer, while using the stronger Llama2-13B yields substantially larger gains. This behavior matches the monotonic trend predicted by our theoretical analysis. We will include the full results in the revised version.
>
> > **Q2: Could the authors quantify the computational cost of trajectory tree construction and MCTS? How scalable is the method to tasks with longer action horizons or larger vocabularies?**
>
> We appreciate the reviewer’s question on computational cost and scalability. Trajectory tree construction is inexpensive in practice because it only processes the weak model’s rollouts and does not require additional model inference. We show our results in the Table below. The MCTS step is similarly lightweight. Both components scale approximately linearly with the action horizon and the number of states in the tree, since all operations are local expansions or pointer traversals. Vocabulary size does not directly affect tree construction or search, since these stages operate purely on already-sampled trajectories rather than enumerating language actions. We will include a short profiling table in the appendix to make these costs explicit and to demonstrate that the method is practical and scalable for larger tasks.
>
> | Task | Tree Build Time (s) | MCTS Rollouts (100) Time (s) |
> |---|---|---|
> | Webshop | 0.41 | 0.23 |
> | Alfworld | 0.68 | 0.38 |
>
> > **Q3: Is there a possibility of iterated W2SG, e.g., bootstrapping progressively stronger models via self-distillation?**
>
> We appreciate the reviewer’s comment. Iterated or multi-round W2SG is certainly a promising extension, and we agree that it could further amplify the benefit of structured weak supervision. In this paper, we deliberately focus on the single-step setting, as it allows a cleaner theoretical treatment and controlled evaluation without introducing the additional complexity that arises in multi-stage bootstrapping. Exploring whether our trajectory-based supervision can support stable iterative W2SG is an exciting future direction, and we plan to investigate it in follow-up work.
>
> Lastly, we want to apologize for the late response. Training the LLMs with SFT takes a long time in the additional experiment.

---

### Author Response · Authors · 2025-12-01
**Rebuttal Summary**

Dear ACs,

We sincerely thank you and all reviewers for your efforts in handling our submission. We would like to summarize the discussion.

 * We have received feedback from Reviewer **yTwq** and we are glad the reviewer is **satisfied** with our response.
     * We have added a paragraph in the introduction to further clarify the relation between our setting and the original W2SG. Basically, our paper has the same goal as the original W2SG and extends the setting to sequential decision-making.
     * We have reported the p-value to judge the significance of the improvement over baselines.
     * We have added a detailed description to clarify the relation between the assumptions and the theorem.
 * After the first round of discussion, Reviewer yTwq raised two additional questions:
      * We have further highlighted the difference between Assumption 3.2 and Assumption 3.3.
      * Reviewer yTwq raised a new question: what if the weak model instead was explicitly misaligned. We remark that both unintentional and intentional errors could be generalized to the strong model. While the reward mechanism in our setting can be considered as a defense strategy, the focus of our paper is to improve informativeness in W2SG.
 * Though Reviewer yTwq was not able to provide further feedback, we have integrated the comments into the revised version.
 * Since Reviewer SFUW and Reviewer jPwv were not able to engage in the discussion, we would like to summarize our new experiments and clarification.
      * Both Reviewer SFUW and Reviewer jPwv raised a question about reliance on the quality of weak models (e.g., what if the weak model is too weak). We conducted additional experiments with two substantially different weak models: Llama2-7B and Llama2-13B, while keeping the strong model fixed as Llama3-8B. The findings confirm the monotonic relationship predicted by our theoretical analysis.  Reliance on the quality of weak models is a shared challenge for all W2SG methods and can be addressed by bootstrapping with intermediate models.
      * We added a formal performance analysis for MCTS following Reviewer SFUW's suggestion.
      * We conducted an extra ablation study to compare with unstructured contrastive pairs following Reviewer SFUW's suggestion. The results further verify the effectiveness of our method.
      * We have reported the computational cost of tree construction and MCTS to answer Reviewer SFUW's question. Compared with the long training time, these costs can be ignored.
      * We have added practical implications of Theorem 1 following Reviewer jPwv's suggestion. Basically, TreeDPO can provide consistent improvement over SFT.
      * We have extended the experiment to Qwen2.5.

 * We have carefully integrated all comments into the revised version. All changes are highlighted in $\textcolor{blue}{\text{blue}}$.

We sincerely express our gratitude to all ACs and reviewers for their dedication to supporting the community.

Best,

Authors

---

### Meta-Review · Area_Chair_ZVBj · 2026-01-03

**Summary:**

This work proposes a new framework for weak-to-strong generalization. It introduces a trajectory tree method that can be used for extraction of DPO pairs. The paper also proposes MCTS for extraction of high-quality paths to be used in imitation. The tree structure can be used to capture shared tree prefixes. The work presents empirical and theoretical evidence of their methodology. For example, by showing that tree DPO can outperform expert level SFT under certain assumptions. The reviewers broadly agreed that this paper presents an interesting take on weak to strong generalization for decision-making tasks. During the rebuttal concerns about the extent of the experimental evaluation were addressed by the authors, in my opinion and in the opinion of the reviewers, with success. Thus I recommend this work for acceptance.

**Reviewer Concerns:**

Some of the reviewer's concerns revolved around the initial limited experimental evaluation of this work, and some presentation issues that the authors addressed successfully.

**Reviewer Scores:**

The lowest score reviewer acknowledged the authors response in a way indicative of their desire to reassess it.

---

### Decision · Program_Chairs · 2026-01-26

Accept (Poster)